# Sequence-dependent base pair stepping dynamics in XPD helicase unwinding

Zhi Qi[1], Robert A Pugh[2,3], Maria Spies[4], Yann R Chemla[1,5]*

[1]Center for Biophysics and Computational Biology, University of Illinois at Urbana-Champaign, Urbana, United States; [2]Institute for Molecular Virology, University of Wisconsin-Madison, Madison, United States; [3]Howard Hughes Medical Institute, University of Winconsin-Madison, Madison, United States; [4]Department of Biochemistry, Carver College of Medicine, University of Iowa, Iowa City, United States; [5]Department of Physics, Center for the Physics of Living Cells, University of Illinois at Urbana-Champaign, Urbana, United States

**Abstract** Helicases couple the chemical energy of ATP hydrolysis to directional translocation along nucleic acids and transient duplex separation. Understanding helicase mechanism requires that the basic physicochemical process of base pair separation be understood. This necessitates monitoring helicase activity directly, at high spatio-temporal resolution. Using optical tweezers with single base pair (bp) resolution, we analyzed DNA unwinding by XPD helicase, a Superfamily 2 (SF2) DNA helicase involved in DNA repair and transcription initiation. We show that monomeric XPD unwinds duplex DNA in 1-bp steps, yet exhibits frequent backsteps and undergoes conformational transitions manifested in 5-bp backward and forward steps. Quantifying the sequence dependence of XPD stepping dynamics with near base pair resolution, we provide the strongest and most direct evidence thus far that forward, single-base pair stepping of a helicase utilizes the spontaneous opening of the duplex. The proposed unwinding mechanism may be a universal feature of DNA helicases that move along DNA phosphodiester backbones.

*For correspondence: ychemla@illinois.edu

**Competing interests:** The authors declare that no competing interests exist.

**Reviewing editor**: Leemor Joshua-Tor, Cold Spring Harbor Laboratory, United States

## Introduction

Helicases are vectorial enzymes that utilize ATP hydrolysis to translocate along single-stranded nucleic acids (NA) and separate the base pairs (bp) of the duplex. The two largest helicase superfamilies, SF1 and SF2, contain a conserved motor core comprised of a nucleotide binding site in the cleft between two RecA-like domains (*Singleton et al., 2007*; *Fairman-Williams et al., 2010*; *Beyer et al., 2013*; *Raney et al., 2013*). Ensemble kinetic and structural studies of SF1 PcrA (*Subramanya et al., 1996*; *Soultanas et al., 1999*; *Dillingham et al., 2000*, *2002*), UvrD (*Ail et al., 1999*; *Fischer et al., 2004*; *Lee and Yang, 2006*) and SF2 nonstructural protein 3 (NS3) (*Tai et al., 1996*; *Kim et al., 1998*; *Pang et al., 2002*; *Levin et al., 2004*; *Tackett et al., 2005*; *Mackintosh et al., 2006*; *Gu and Rice, 2010*) suggest these domains move as a ratchet-like inchworm, whereby the helicase translocates along single-stranded NA by 1 nucleotide (nt) during each ATP binding and hydrolysis cycle. Despite these findings, the mechanism of base pair separation has remained elusive. Ensemble and single-molecule kinetic studies of RecBC (*Bianco and Kowalczykowski, 2000*; *Lucius et al., 2004*) and NS3 (*Dumont et al., 2006*; *Myong et al., 2007*) have reported unwinding in steps nested within larger steps. A more recent single-molecule study of NS3 helicase confirmed a single-base pair unwinding step size but also reported 1/2-bp steps, which were attributed to asynchronous release of the unwound NA (*Cheng et al., 2011*). Unwinding mechanisms where the helicase unwinds duplex as a monomeric inchworm (*Velankar et al., 1999*; *Singleton et al., 2007*) or requires helicase oligomerization (*Lohman et al., 2008*) have been proposed. Lastly, the mechanism of base pair separation—whether achieved by

**eLife digest** During many cellular processes, the double helix must be transiently unwound so that the enzymes responsible for maintaining the genome can access the two strands. During DNA synthesis, for instance, the two strands of DNA are first separated and then used as templates for the production of new strands. The role of destabilizing, separating and unwinding the double helix falls to enzymes known as DNA helicases.

Helicases are also involved in separating strands of nucleic acids during myriad other cellular processes, including DNA repair, transcription and translation. While the functions of helicases are clear, the precise mechanisms by which they unwind DNA are not.

Here, Qi et al. have investigated the mechanism of a helicase called XPD, which is involved in DNA repair and the initiation of transcription of DNA into RNA. Using optical tweezers—in which a laser beam is used to exert extremely small forces on a single DNA molecule—they followed the activity of individual molecules of XPD as they unwound DNA with base pair resolution.

Qi et al. observed that the helicase unwinds DNA strands 1 base pair at a time, but that it sometimes moves backwards by 1 base pair and at other times makes 5 base pair backward and forward steps. The frequency of these backwards steps depends on the availability of ATP, and the sequence of the DNA. Due to the high resolution of the data, Qi et al. were able to correlate these stepping dynamics with the DNA sequence with base pair level accuracy. While some helicases actively separate the strands, using energy derived from ATP to break the hydrogen bonds between pairs of bases, Qi et al. showed that XPD appears to take advantage of momentary separations that arise spontaneously between base pairs.

As well as providing insights into the role of XPD in DNA repair and transcription, the work of Qi et al. presents a method that could be used to explore the mechanisms of other helicases. Given that the unwinding mechanism described here is likely to be a universal feature of enzymes related to XPD, the current work could shed light on a number of other cellular processes involving XPD-like helicases, such as homologous DNA recombination, inter-strand cross-link repair, and accurate chromosome segregation.

helicase destabilizing base pairs directly (*Johnson et al., 2007*) or by rectifying thermal breathing of the duplex (*Lionnet et al., 2007*)—remains disputed. Previous studies investigating this question by varying NA sequence (*Cheng et al., 2007*; *Johnson et al., 2007*; *Manosas et al., 2010*) have been limited, as assumptions on helicase step size, slippage, and backstepping have prevented definitive statements (*Manosas et al., 2010*). These studies have lacked the resolution to assess stepping dynamics directly. Distinguishing between mechanisms requires interrogating helicase activity at the single-protein level, and with high spatio-temporal resolution.

Here, we applied high-resolution dual-trap optical tweezers (*Abbondanzieri et al., 2005*; *Moffitt et al., 2009*) to determine the mechanism underlying activity of XPD helicase from *Ferroplasma acidarmanus*. This approach allowed us to monitor individual XPD molecules acting on a dsDNA hairpin with base pair resolution. XPD is a prototypical 5′–3′ translocating SF2 helicase (SF2B) (*Singleton et al., 2007*; *Wolski et al., 2010*). It consists of four structural domains: helicase domains 1 and 2 (HD1 and HD2), which contain all helicase signature motifs important for coupling ATP binding and hydrolysis to directional translocation along ssDNA, and two unique modular domains (ARCH and FeS) inserted in the conserved motor core (*Figure 1A*; *Fan et al., 2008*; *Liu et al., 2008*; *Wolski et al., 2008*). All known SF2B enzymes participate in DNA repair or support replication and, therefore, are important for maintenance of genomic integrity (*White, 2009*; *Wu et al., 2009*). Human XPD is a player in nucleotide excision repair and transcription initiation (*Egly and Coin, 2011*). Related FeS-containing helicases FANCJ, RTEL and CHLR1 share structural organization and likely the same unwinding mechanism as XPD (*Gupta et al., 2007*; *White and Dillingham, 2012*). The features of the DNA unwinding mechanism determined for XPD may thus be broadly relevant to SF2 helicases, which all make contacts with the phosphodiester backbone of nucleic acids.

The high sensitivity of our experimental approach allowed us to determine that monomers of XPD helicase unwind DNA in 1 base pair steps. We further established that XPD is an inefficient helicase with low processivity that displays repetitive attempts at unwinding duplex DNA. In contrast to other

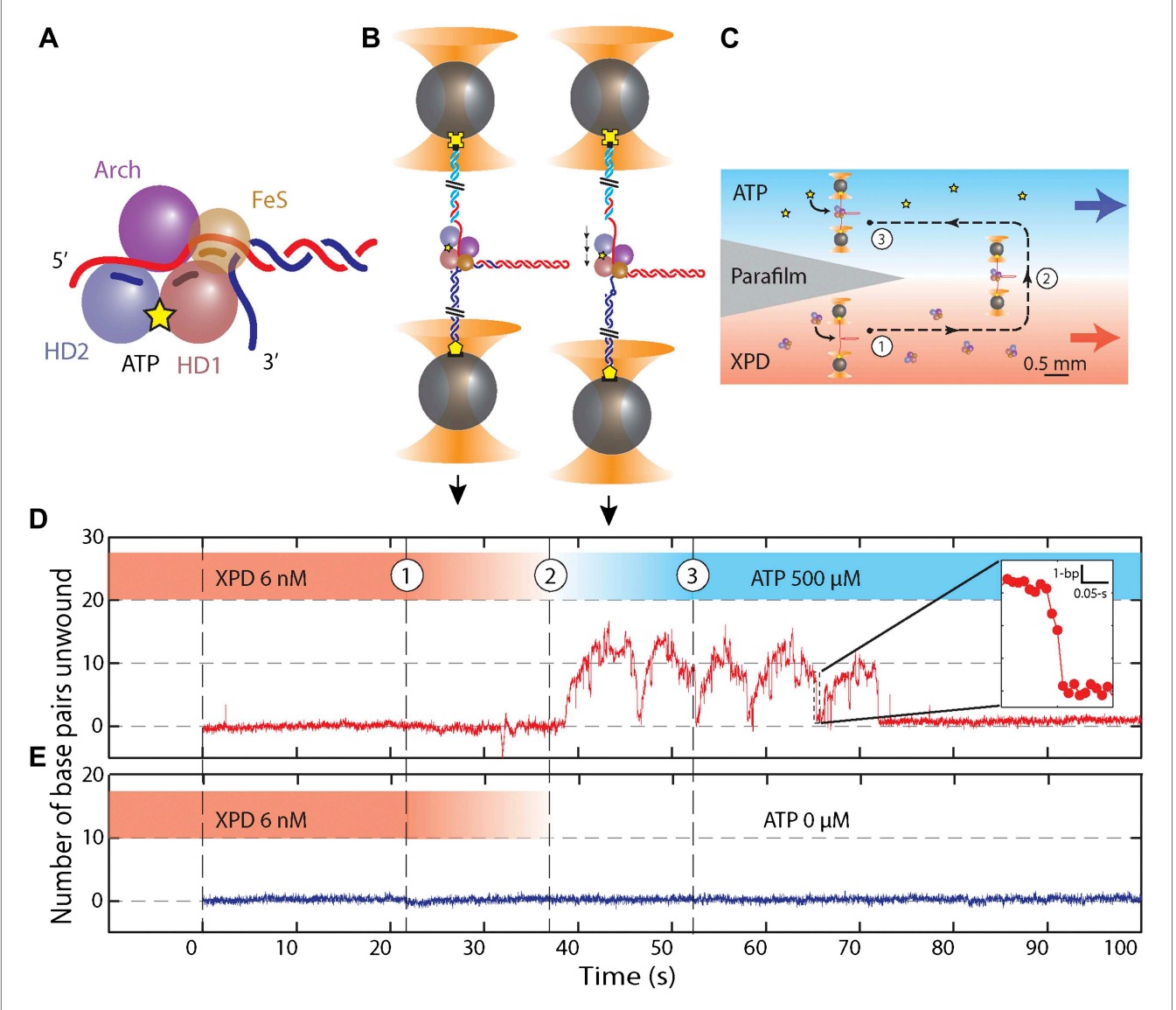

**Figure 1**. Unwinding activity of individual XPD helicase. (**A**) Schematic of XPD helicase with DNA. The 5'–3' helicase consists of two RecA-like domains (HD1, salmon; HD2, blue) forming the Rad3 motor core that hydrolyzes ATP (yellow star), an ARCH domain (purple), and a FeS cluster (brown) that belongs to HD1. (**B**) Schematic of the hairpin-unwinding assay (**Dumont et al., 2006**). A single construct was tethered between two trapped beads via biotin-streptavidin (yellow cross) and digoxigenin-antibody (yellow pentagon) linkages. A single helicase bound to the 5' $(dT)_{10}$ ssDNA loading site. Unwinding of the 89-bp hairpin was monitored from the change in end-to-end extension of the molecule under a constant tension of 12 pN. (**C**) Experimental layout. A laminar flow cell created two adjacent streams of buffer containing XPD only (red; [XPD] = 6 nM) and ATP only (blue; [ATP] = 0–500 µM), respectively. During an experiment, a tether was incubated at position (1) for 20 s, and then moved along the dashed-line path across the stream interface (2) and into the ATP-rich stream (3). (**D**) ATP-dependent single-helicase activity corresponding to (**C**). Inset highlights one XPD backslide. (**E**) Control experiments in an ATP-free stream displayed no unwinding activity.

The following figure supplements are available for figure 1:

**Figure supplement 1**. DNA hairpin substrate design.

**Figure supplement 2**. Force extension curves of DNA hairpin.

**Figure supplement 3**. Laminar flow cell design.

helicases, backstepping plays a prominent role in XPD helicase mechanism. XPD exhibited frequent single-base pair backsteps, which depended on ATP concentration and DNA sequence, and larger backstepping events that likely corresponded to conformational rearrangements of the protein–DNA complex. Our high resolution assay allowed us, for the first time, to correlate with near base pair registration the effect of DNA sequence on XPD's stepping dynamics. We show that XPD unwinding relies strongly on spontaneous opening of the base pairs ahead, and that this alone accounts for the observed low processivity. Based on our measured kinetic parameters we propose a mechanism for XPD unwinding and speculate on its applicability to other biomedically important SF2 helicases.

## Results

### Single XPD helicase exhibits repetitive, non-processive unwinding bursts

We monitored XPD-mediated unwinding of an 89-bp hairpin containing a random DNA sequence (see 'Materials and methods'; *Figure 1B* and *Figure 1—figure supplement 1*). A 10-dT single-stranded DNA (ssDNA) binding site at the 5′ end of the hairpin allowed loading of a single XPD molecule (*Kim et al., 1998*; *Singleton et al., 2007*). In the presence of ATP, XPD unwinds the hairpin, releasing 2 ssDNA nucleotides (nt) for every base pair unwound (*Dumont et al., 2006*; *Figure 1B*). Unwinding was measured at a constant tension of 12 pN, insufficient to mechanically unfold the hairpin (see 'Materials and methods'; *Figure 1—figure supplement 2*). To prevent hairpin unwinding by multiple helicases, we developed a protocol in which a single protein could be loaded onto the construct and unwinding monitored in the absence of other proteins in solution. We used a laminar flow chamber (*Brewer and Bianco, 2008*; see 'Materials and methods'; *Figure 1—figure supplement 3*) with two parallel streams of buffer containing ATP (0–500 µM) and 6 nM XPD, respectively (blue and red streams, *Figure 1C*). A single tethered hairpin was incubated in the XPD stream (position 1, *Figure 1C*) for 20 s to allow the protein to bind. Hairpin unwinding could be detected soon after moving the tether into the ATP stream (position 2, *Figure 1C,D*). Control experiments with no ATP showed no unwinding of the hairpin to within 0.4 bp (representing 1 SD in the hairpin extension noise; N = 19 traces, *Figure 1E*).

*Figure 1D* displays the activity typically observed from a single XPD helicase (see below). Several features are notable. First, the protein exhibited repetitive unwinding 'bursts' during which the hairpin gradually unfolded, then suddenly and completely re-annealed. While unwinding was gradual, re-annealing occurred in large (−6.3 ± 1.6 bp, mean ± SD; N = 144 bursts) and sudden backward jumps (*Figure 1D*, inset). The re-annealing rate was high (>100 bp/s; N = 144 bursts), much faster than the translocation speed of 13 ± 2 nt/s (*Honda et al., 2009*) and ATP independent, inconsistent with helicase translocation on the opposite strand, as observed in other systems (*Dessinges et al., 2004*). These observations suggest a mechanism of repetitive unwinding in which XPD can backslide by several base pairs without dissociating from its DNA substrate. Structures and biochemical studies suggest that the ARCH domain of XPD can encircle the translocating DNA strand during unwinding (*Liu et al., 2008*; *Wolski et al., 2008*; *Kuper and Kisker, 2012*; *Kuper et al., 2012*; *Pugh et al., 2012*), providing a potential structural basis for backsliding without dissociation (see 'Discussion'). Although XPD's behavior is reminiscent of the 'repetitive shuttling' previously reported for the SF1 helicase Rep (*Myong et al., 2005*), some key differences distinguish the two. First, Rep helicase shuttling was detected during translocation on ssDNA, not duplex unwinding. Second, shuttling in Rep is mediated by formation of a transient DNA loop that allows rapid 'snapback' to its initial position on DNA. Our data disfavors such a mechanism, as the re-annealing usually occurred in multiple jumps, rather than a single snapback. Furthermore, formation of a ~12-bp loop is unlikely at the applied tension. As discussed below, our analysis of stepping kinetics further disfavor a looping mechanism in XPD. After several unwinding bursts and backslides (*Table 1*), activity suddenly ceased, indicating protein dissociation. In none of our measurements (N = 363 traces) did activity return once a dissociation event was observed. The second notable feature is the low processivity of XPD helicase (*Figure 1D*). Although the full hairpin measured 89 bp, on average only 12 ± 3 bp (mean ± SD; *Table 1*) were unwound during each burst.

### Demonstration of single-protein unwinding activity

We performed several control experiments to ensure that the activity depicted in *Figure 1D* was obtained from a single protein. First we designed constructs containing ssDNA binding sites of varying

**Table 1.** Summary of experimental data*

| Sequence | 1 | 1 | 1 | 1 | 1 | 1 | 2 |
|---|---|---|---|---|---|---|---|
| ATP [μM] | 6.25 | 12.5 | 25 | 50 | 250 | 500 | 12.5 |
| No. of traces | 5 | 50 | 31 | 13 | 12 | 40 | 11 |
| Total No. of bursts | 13 | 118 | 82 | 40 | 41 | 144 | 28 |
| Burst / trace | 2.0 ± 0.4 | 2.0 ± 0.3 | 2.6 ± 0.3 | 3.1 ± 0.4 | 3.7 ± 0.6 | 3.6 ± 0.5 | 4.7 ± 0.9 |
| Mean processivity (bp)† | 11 ± 2 | 11 ± 2 | 12 ± 3 | 13 ± 2 | 12 ± 3 | 12 ± 3 | 21 ± 6 |
| No. of bursts for PWD (%)‡ | 8 (61.5) | 58 (49.2) | 36 (43.9) | 11 (27.5) | 4 (9.8) | 48 (33.3) | 12(42.9) |
| No. of bursts for stepfitting§ | 11 | 64 | 70 | 17 | 16 | 126 | 12 |
| Total No. of steps | 186 | 827 | 1022 | 179 | 207 | 1612 | 559 |
| Non-fitting points (%) | 4.7 | 3.1 | 5.7 | 4.0 | 6.3 | 6.2 | 5.2 |
| No. of 1-bp steps# | 115 | 522 | 646 | 94 | 101 | 845 | 332 |
| Dwell time (ms)¶ | 370 ± 35 | 274 ± 12 | 212 ± 8 | 201 ± 21 | 183 ± 18 | 178 ± 6 | 281 ± 14 |
| Step size (bp)** | 1.1 ± 0.4 | 1.1 ± 0.5 | 1.1 ± 0.5 | 0.8 ± 0.2 | 1.1 ± 0.5 | 1.0 ± 0.4 | 0.9 ± 0.3 |
| No. of +1/+1 bp step pairs# | 54 | 271 | 351 | 62 | 69 | 643 | 169 |
| No. of +1/−1 bp step pairs# | 23 | 104 | 129 | 13 | 11 | 88 | 74 |
| No. of −1/+1 bp step pairs# | 26 | 99 | 117 | 7 | 15 | 87 | 62 |
| No. of −1/−1 bp step pairs# | 12 | 48 | 49 | 12 | 6 | 27 | 27 |
| No. of large steps# | 17 | 72 | 70 | 22 | 23 | 155 | 18 |
| Dwell time (ms)¶ | 158 ± 38 | 150 ± 18 | 149 ± 18 | 162 ± 34 | 139 ± 29 | 127 ± 10 | 132 ± 29 |
| Backstep size (bp)** | −4.5 ± 0.8 | −4.6 ± 1.1 | −4.5 ± 0.7 | −4.7 ± 0.8 | −5.0 ± 1.0 | −4.6 ± 0.9 | −3.8 ± 0.6 |
| Forward step size (bp)** | 4.0 ± 1.0 | 4.3 ± 0.9 | 4.5 ± 0.9 | 4.7 ± 1.1 | 4.5 ± 0.8 | 4.4 ± 0.9 | 3.6 ± 0.4 |

*All data from hairpin "sequence 1" containing a 10-nt helicase binding site (see 'Materials and methods') at a 12 pN force.

†Errors are SEM except where otherwise noted.

‡Processivity is defined as the maximum number of base pairs unwound (mean ± SD).

§A selection of bursts with low noise properties was used for PWD and step-fitting analysis (see 'Materials and methods').

#Steps with size $0 \leq d \leq 2$ were scored as +1-bp steps; $-2 \leq d \leq 0$ as −1-bp backsteps; $3 \leq d \leq 6$ as large steps; $-6 \leq d \leq -3$ as large backsteps.

¶Mean dwell times were determined by fitting dwell time histograms to an exponential (mean ± 95% confidence intervals).

**Mean step sizes over the region of interest were determined by fitting step size histograms to a Gaussian (mean ± 95% confidence intervals).

length (see 'Materials and methods'; *Figure 1—figure supplement 1*). Although the minimal footprint enabling efficient loading of a single XPD is unknown, we estimate it to be 6–8 nt based on structural information on SF2 helicase complexes with nucleic acids (*Kim et al., 1998*; *Buttner et al., 2007*; *Singleton et al., 2007*; *Gu and Rice, 2010*) and the predicted binding mode of XPD (*Kuper et al., 2012*; *Pugh et al., 2012*). No unwinding activity was detected on the constructs containing a ssDNA binding site smaller than the expected XPD footprint (0-dT and 3-dT; *Figure 2A,B*), indicating that XPD helicase could not initiate unwinding of these substrates. In contrast, unwinding activity was readily detected (*Figure 2C,D*) when the binding sites were ≥10 nt. Interestingly, we observed two types of unwinding behaviors for long binding sites: a 'low-processivity' activity (*Figures 1D and 2C*), in which ~12 bp were unwound repetitively; and a 'processive' activity (*Figure 2D*), in which the 89-bp hairpin was completely unwound. The latter was observed only with constructs containing binding

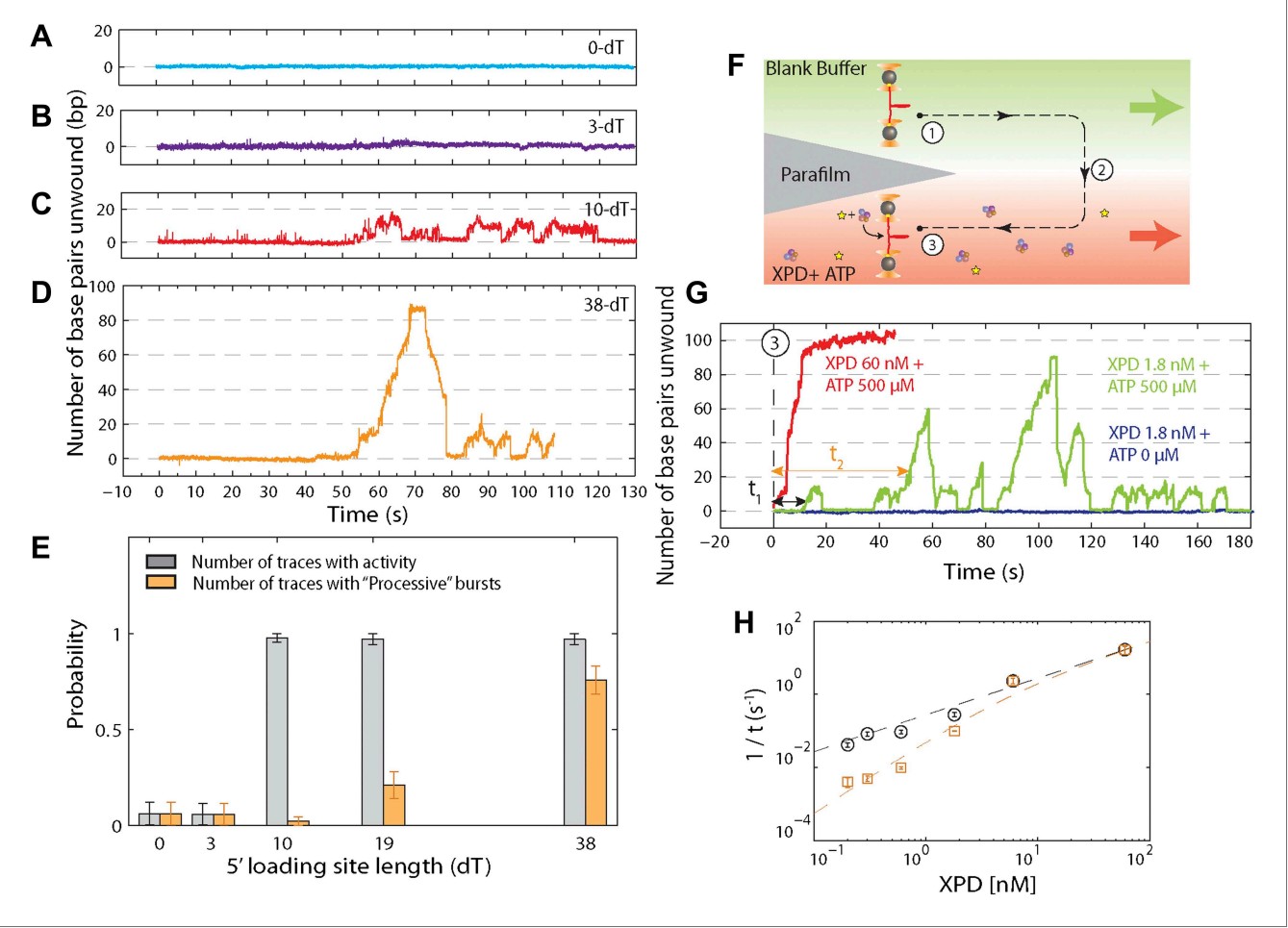

**Figure 2**. Effect of 5' loading site length and XPD concentration on activity. (**A**)–(**D**), Representative XPD unwinding traces for different helicase loading site lengths: 0-dT, 3-dT, 10-dT, and 38-dT. The traces in (**A**) and (**B**) show no unwinding activity, which we defined as hairpin unfolding by >5 bp. In contrast, the traces in (**C**) and (**D**) exhibit activity; we defined this activity as low-processivity (**C**) if the 89-bp hairpin was not completely unwound, or processive (**D**) if it was. Traces were occasionally seen to revert from processive to low-processivity activity (**D**), potentially indicating dissociation of one of the cooperating XPD molecules while leaving the other remaining. (**E**) Fraction of active traces (gray bars) and processive traces (orange bars) vs 5' loading site length: 0-dT, 3-dT, 10-dT, 19-dT and 38-dT (N = 14, 15, 40, 31, and 31 tethers, respectively). No activity was detected when the loading site was <10 nt. Processive activity was detected only when the binding site was ≥19 nt, long enough to accommodate two or more XPD helicases. Probabilities were calculated from the Laplace estimator ($n_{success}$ + 1)/($N_{trial}$ + 2). Error bars throughout denote SD. (**F**) Experimental layout for XPD titration measurement. The laminar flow cell was used to create two adjacent streams of buffer containing XPD mixed with ATP (red; [XPD] = 0.2–60 nM and [ATP] = 500 μM) and blank buffer only (green), respectively. A single tether was formed at position (1), and then moved along the dashed-line path across the stream interface (2) and into the XPD + ATP-rich stream (3). (**G**) Representative XPD-dependent helicase activity. Unwinding traces in the presence of [XPD] = 60 nM (red) and 1.8 nM (green) are shown. The waiting times $\tau_1$ (and $\tau_2$) were defined as the total time elapsed from entering the XPD + ATP-rich stream until the initial low-processivity (and processive) unwinding activity was detected. Control experiments in which the tether was moved into an ATP-free stream with 1.8 nM XPD (navy) displayed no unwinding activity. (**H**) 1/$\tau$ vs XPD concentration. 1/$\tau_1$ (black) and 1/$\tau_2$ (orange) were fitted to the models described in the main text ([XPD] = 0.2, 0.3, 0.6, 1.8, 6.0, and 60 nM, N = 27, 59, 71, 56, 16, and 13 binding events, respectively). Error bars throughout denote SEM.
The following figure supplements are available for figure 2:

**Figure supplement 1**. Dependence of unwinding rates of single XPD and multiple XPDs on ATP concentration.

sites >10 nt that were expected to accommodate two or more XPD monomers (19-dT and 38-dT; *Figure 2E*). This result suggests that individual motors can unwind ~12 bp while multiple motors may cooperate to increase unwinding processivity or generate forces sufficient for overcoming difficult sequences (*Eoff and Raney, 2010*). Traces were occasionally seen to revert from processive to low-processivity activity (*Figure 2D*), potentially indicating dissociation of one of the cooperating

XPD molecules while leaving the other engaged. Interestingly, although processivity increased with multiple motors, the mean unwinding rate remained unchanged (*Figure 2—figure supplement 1*), as similarly observed with NS3 helicase (*Levin et al., 2004*; *Tackett et al., 2005*).

In the second set of control experiments we monitored unwinding in the presence of varying XPD concentration. To this end, we used a different stream configuration in the laminar flow chamber: the upper stream contained buffer only and the lower stream buffer with both XPD (0.2–60 nM) and ATP (500 µM; *Figure 2F*). Provided the XPD concentration was low (≤1.8 nM, green line, *Figure 2G*), most of the activity observed was low-processivity, similar to that shown in *Figure 1D*. For XPD concentrations ≥6 nM, only processive activity could be detected (red line, *Figure 2G*), consistent with our interpretation that this activity resulted from multiple proteins. Control measurements performed in the absence of ATP showed no activity (navy line, *Figure 2G*). We measured the waiting time from the moment the tether was moved to the lower stream containing XPD and ATP until the detection of the first unwinding activity ($\tau_1$) and the first processive unwinding activity ($\tau_2$). The rate $1/\tau_1$ for first unwinding varied linearly with XPD concentration (black circles, *Figure 2H*), indicating that $\tau_1$ corresponds to the time for a single XPD protein to bind to the DNA. (We note that there has been no evidence for XPD dimerization in solution under buffer conditions similar to ours; *Pugh et al., 2008b*). The data were fitted to $1/\tau_1 = k_{on}^{XPD} [XPD]$ yielding the association rate constant $k_{on}^{XPD} = 0.28 \pm 0.01 \text{ s}^{-1} \text{ nM}^{-1}$ (mean ± SD). In contrast, the rate $1/\tau_2$ for first processive unwinding depended on a higher power of the XPD concentration (orange squares, *Figure 2H*), indicating a requirement for multiple proteins. We fitted the data to a simple model in which the rate for first processive unwinding corresponds to binding of a second XPD to the DNA. Thus, $1/\tau_2 = p_{XPD} k_{on}^{XPD2} [XPD]$, where $p_{XPD}$ is the probability that one XPD helicase is already bound to the DNA when the second binds and $k_{on}^{XPD2}$ is the association rate constant for the second helicase, $p_{XPD} = \dfrac{[XPD]}{[XPD] + k_{off}^{XPD} / k_{on}^{XPD}}$, where $k_{off}^{XPD}$ is the dissociation rate constant. The data were fitted to this expression, yielding equilibrium dissociation constant $k_d = k_{off}^{XPD} / k_{on}^{XPD} = 5 \text{nM}$ (similar to that observed in bulk studies; *Pugh et al., 2008a*), and $k_{on}^{XPD2} \approx k_{on}^{XPD}$. The data are consistent with low-processivity behavior being due to single-XPD activity, and processive activity from cooperative action of multiple (at minimum, two) monomers.

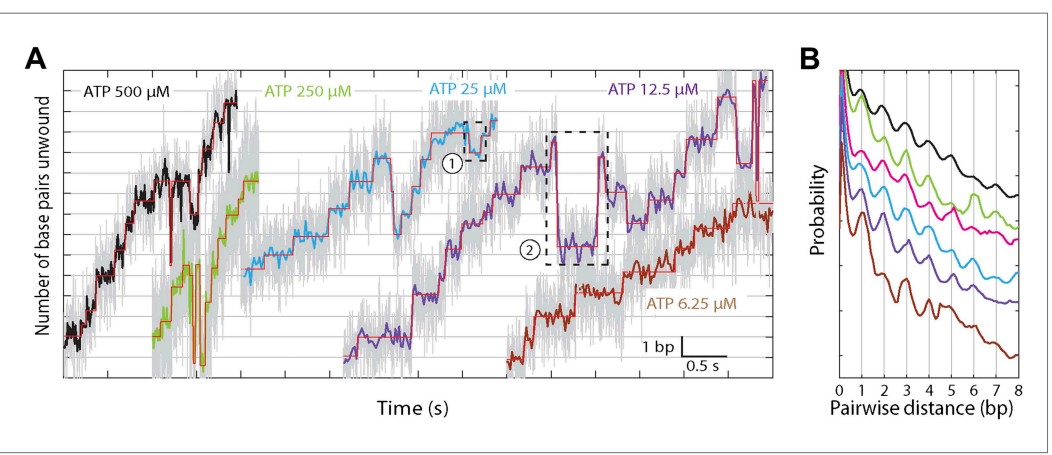

**Figure 3**. XPD stepping behavior. (**A**) Representative traces of XPD unwinding at 500, 250, 25, 12.5, and 6.25 µM ATP (black, light green, cyan, purple, and brown respectively; data filtered and decimated to 100 Hz ([ATP] = 500, 250 µM) and 50 Hz (25, 12.5, 6.25 µM). Raw data in gray acquired at 1 kHz). A step-fitting algorithm (*Kerssemakers et al., 2006*) was used to fit the data (red). Dashed rectangle (1) highlights a 1-bp backstep; dashed rectangle (2) highlights a large 5-bp backward and forward step event. (**B**) PWD analysis for selected traces at all ATP concentrations. The color map is the same as (**A**), with pink for 50 µM ATP.

The following figure supplements are available for figure 3:

**Figure supplement 1**. XPD stepping dynamics for alternate sequences and forces.

## XPD helicase unwinding occurs in 1-base pair steps with frequent backsteps

The base pair resolution of our assay allowed us to investigate the detailed mechanism behind the observed repetitive, low-processivity activity. As shown in example traces in *Figure 3A*, XPD unwound DNA in discrete steps but did not appear to step exclusively along one direction. Based on a pairwise distribution (PWD) analysis (*Abbondanzieri et al., 2005*; *Dumont et al., 2006*; *Moffitt et al., 2009*), we determined an elemental step size of 1 bp across all ATP concentrations (*Figure 3B*; see 'Materials and methods'; *Table 1*). Unwinding activity was also measured on two alternate hairpin sequences and across a range of tensions (see 'Materials and methods'; *Figure 1—figure supplement 2*). PWD analysis of these unwinding data revealed the same 1-bp step size throughout (*Figure 3—figure supplement 1*), indicating that our determination of the step size is robust, independent of sequence or tension.

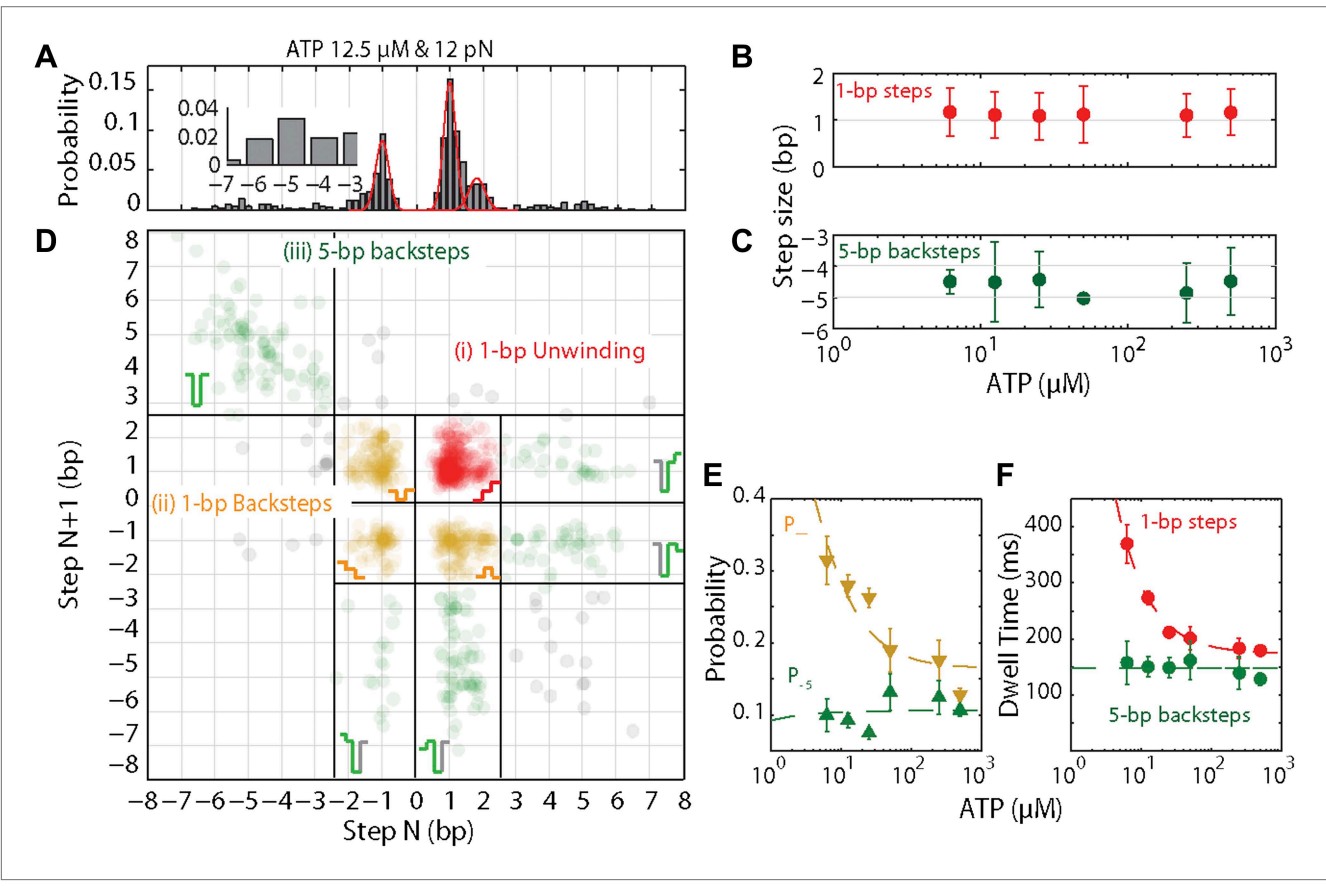

**Figure 4**. XPD stepping dynamics and dependence on ATP. (**A**) Representative histogram of step sizes for 12.5 μM ATP, with Gaussian fit (red). Step sizes were determined from the step-fitting algorithm. Inset highlights the larger backsteps. (**B**) and (**C**) Mean step size for elemental 1-bp steps (**B**) and large 5-bp backsteps (**C**) vs ATP concentration. Means were determined from the Gaussian fits of the step size distribution obtained from the step finding algorithm (**A**). Error bars throughout denote 95% confidence intervals from Gaussian fit. (**D**) Representative scatter plot of step pairs for 12.5 μM ATP. Each data point represents the step sizes of every adjacent pair of steps. Three regions are highlighted: (i) Step pairs with consecutive 1-bp unwinding steps (red); (ii) pairs with 1-bp backsteps (orange); and (iii) pairs with larger (~5 bp) backsteps (dark green). Grey points represent the small fraction of step pairs that did not fit in the categorization above. (**E**) Probabilities for taking 1-bp backsteps, $p_-$ (orange), and for taking larger backsteps, $p_{-5}$ (dark green), vs ATP concentration. Probabilities were calculated from the Laplace estimator ($n_{success}$ + 1)/($N_{trial}$ + 2). Error bars denote SD. (**F**) Dwell times for 1-bp steps vs ATP concentration. Error bars denote SEM. Dashed lines in (**E**) and (**F**) represent a global fit to the kinetic model described in the text and 'Materials and methods'. The reduced $\chi$ value for the global fit to the data in panels E and F was 3.2. Experimental details are summarized in *Table 1*.

The following figure supplements are available for figure 4:

**Figure supplement 1**. Dependence of large 5-bp backsteps on tension.

**Figure supplement 2**. Distribution of dwell times for large 5-bp backsteps.

In addition to the PWD analysis, we also applied a step-fitting algorithm (*Kerssemakers et al., 2006*) to fit the unwinding traces (red lines, *Figure 3A*). The histogram of step sizes from this analysis (*Figure 4A*) corroborates the 1-bp elemental step size across the range of ATP concentrations assayed (*Figure 4B*). Inspection of individual traces reveals that not all steps measured an exact integer number of base pairs. These events, which we attribute to measurement noise, are responsible for the width of the peaks in the PWD and in the step size histogram, but were not significant enough to disrupt the 1-bp pattern. In particular, our analyses provide little evidence for a statistically significant 0.5-bp step size, as recently reported for NS3 helicase and interpreted as a manifestation of transient DNA looping in that system (*Cheng et al., 2011*).

In addition to unwinding in 1-bp steps, the protein exhibited frequent backsteps (dashed rectangle 1 in *Figure 3A*) and occasional large (~5 bp) steps backward and forward (dashed rectangle 2 in *Figure 3A*). Our step-fitting algorithm reveals that the majority of backsteps measure 1 bp while the less common large backsteps span −4.6 ± 0.9 bp (mean ± SD). In both cases the step sizes were independent of ATP (*Figure 4B,C*). Both types of events are distinct from the backsliding described above. Single-base pair backsteps have so far only been observed in high-resolution optical trap measurements of NS3 helicase, and appear rare (*Cheng et al., 2011*). In contrast, backsteps are very frequent for XPD. Moreover, the large 5-bp steps backward and forward exhibited by XPD appear to be a new feature of helicase stepping mechanics that, to our knowledge, has not been reported before.

To capture the complexity of observed backstepping behavior, we analyzed steps in pairs. *Figure 4D* shows a scatter plot of the step sizes for all adjacent pairs of steps detected at 12.5 μM ATP and 12 pN. The different types of stepping behavior are evident. Red data points represent consecutive 1-bp forward steps, orange points depict step pairs with one or two 1-bp backsteps, and green points depict pairs with a large 5-bp backward or forward step. (We also observed a small fraction of 2-bp steps, which we attributed to closely spaced 1-bp steps missed by the step-finding algorithm.) Nearly all step pairs plotted corresponded to combinations of forward and backward 1-bp and ~5-bp steps; the small fraction (~5%, across all ATP concentrations, *Table 1*) of events that did not fit categorization are represented by the gray points. We performed this analysis across the range of ATP concentrations assayed and determined the probability, $p_-$, of a 1-bp backstep relative to all 1-bp steps. $p_-$ varied inversely with ATP, but importantly remained significant (~15%) at saturating ATP concentrations (*Figure 4E*). We found that probabilities for pairs of adjacent 1-bp steps were simply products of individual probabilities (i.e., the probability for two consecutive backsteps was $p_-^2$), indicating that the direction of each 1-bp step was independent of the preceding step. In contrast, the large 5-bp steps displayed a strong correlation to past behavior; 5-bp backsteps were almost always followed by equal-size forward steps (−4.6 vs 4.4 ± 0.9 bp, mean ± SD; [ATP] = 500 μM and $F$ = 12 pN; *Table 1*). This feature distinguishes the 5-bp backsteps from the backsliding events described above, as the latter were always followed by a 1-bp step. The pairing of 5-bp backsteps with 5-bp forward steps suggests that XPD kept track of its position along the hairpin after the initial backstep. The probability for 5-bp backsteps $p_{-5}$ was low (5–10%) and depended weakly on ATP. Measurements across a range of tensions on alternate hairpin sequences revealed that the frequency of 5-bp steps depends inversely on force (*Figure 4—figure supplement 1*; dependence on sequence is discussed further below). In addition to probabilities, we also determined the dwell times between steps. As shown in *Figure 4F*, the dwell times $\tau$ between all 1-bp steps (forward or backward) exhibited Michaelis-Menten-like behavior. In contrast, dwell times between 5-bp backsteps and subsequent large forward steps were independent of ATP (*Figure 4F*). Distributions of those dwell times were exponential, indicating a single rate-limiting kinetic step (*Figure 4—figure supplement 2*).

## Stepping kinetics are highly sequence-dependent

To understand the mechanism underlying forward- and backstepping by XPD, we next investigated the role of DNA hairpin sequence (*Figure 5A*). The histogram of the unwinding run lengths or processivity (defined here as the maximum number of base pairs unwound per unwinding burst; *Figure 5B*) displays a sharp peak at the 12th base pair, immediately preceding a patch in the hairpin with high G-C content, suggestive of a potential correlation. To quantify this correlation, we determined the probability at each position $n$ along the hairpin, $p_{open}(n,F)$, that one or more base pairs downstream of the hairpin fork would open at the applied tension $F$ (*Johnson et al., 2007*; *Huguet et al., 2010*) ('Materials and methods'). $p_{open}(n,F)$ is correlated with sequence AT% (*Figure 5—figure supplement 1*),

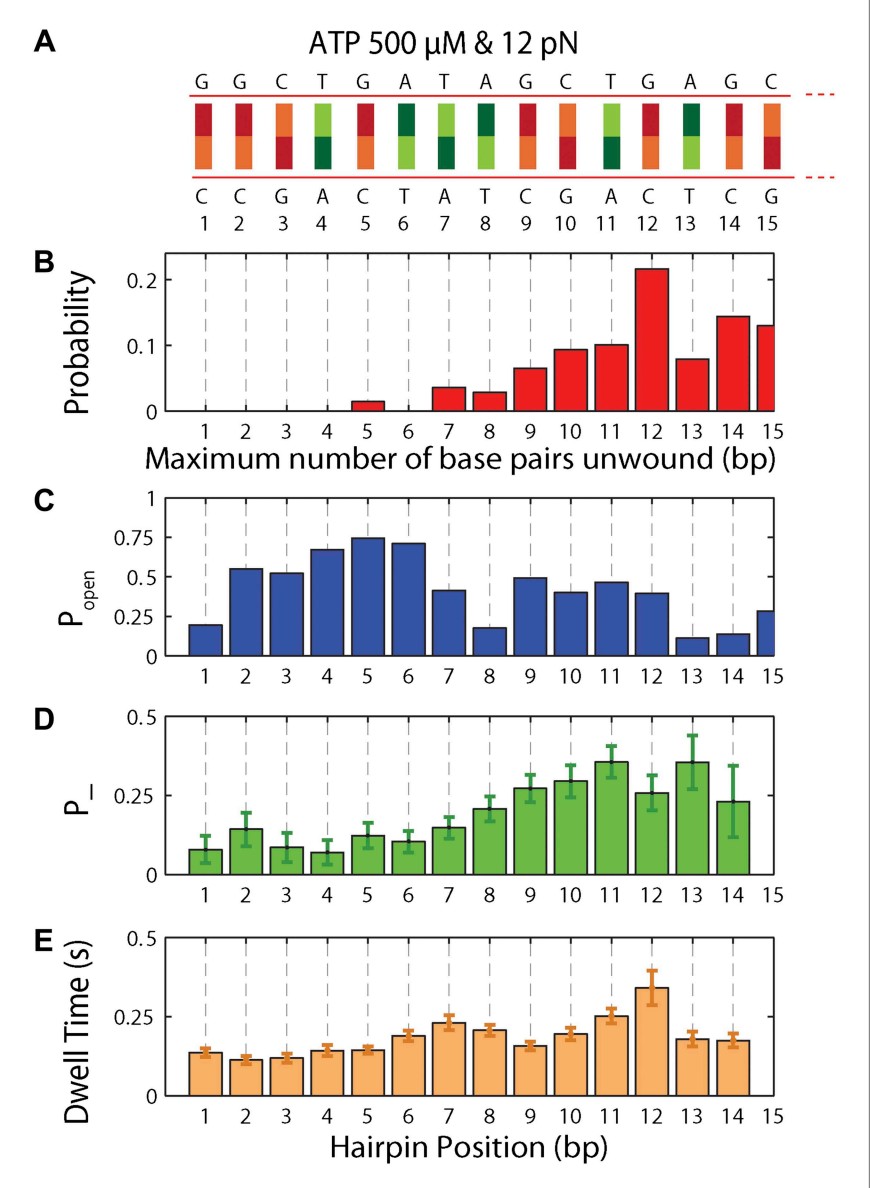

**Figure 5**. Sequence dependence of XPD processivity, backstepping probability, and dwell time. (**A**) DNA hairpin sequence from base pair 1 to 15. (**B**) Histogram of the processivity (N = 144 bursts). (**C**) Hairpin fork opening probability $p_{open}$ vs hairpin position. (**D**) Mean backstepping probability vs hairpin position (N = 1612 total steps). (**E**) Mean dwell time $\tau$ vs hairpin position. Data for (**B**), (**D**) and (**E**) were obtained at saturating ATP conditions (500 μM), under which condition translocation is rate-limiting.

The following figure supplements are available for figure 5:

**Figure supplement 1**. Correlation between *AT%* and $p_{open}$.

**Figure supplement 2**. Effect of hairpin sequence on XPD helicase processivity.

**Figure supplement 3**. Representative XPD unwinding bursts for different sequences and forces.

**Figure supplement 4**. Dependence of large 5-bp backsteps on sequence.

**Figure supplement 5**. Effect of tension on XPD helicase processivity.

a parameter more typically used to describe DNA sequences. For our purposes, we use $p_{open}$ as it quantifies directly the probability of thermal fluctuations opening the hairpin, based on its sequence and the applied tension. *Figure 5C* shows that the position where the majority of unwinding bursts stalled matches the minimum in $p_{open}$. This strong sequence dependence is further substantiated in measurements with two alternate hairpin sequences (see 'Materials and methods'; *Figure 1—figure supplement 2*). For all three hairpins, we observed a strong correlation between the processivity and positions of minimum $p_{open}$ (*Figure 5—figure supplements 2 and 3*).

The high-resolution data further allowed us to correlate the stepping statistics to XPD's position along the hairpin with base pair registration accuracy. *Figure 5D,E* display $p_-$ and $\tau$ as a function of position at saturating ATP. In both cases, the data suggest an anticorrelation with $p_{open}$. (In comparison, the 5-bp backstepping probability $p_{-5}$ displayed hardly any dependence on sequence (*Figure 5—figure supplement 4*)). We related $p_-$ and $\tau$ to the rate constants describing forward and reverse motion, $k_F$ and $k_R$ (*Norstrom et al., 2010*; see 'Materials and methods'). The probability of backstepping is the kinetic competition between a reverse vs a forward step; that is, $p_- = k_R/(k_R + k_F)$ while the dwell time is the inverse of the sum of the two microscopic rate constants: $\tau = 1/(k_R + k_F)$. Combining the two expressions, we determined $k_F$ and $k_R$. *Figure 6A–C* display $p_-$, $k_F$ and $k_R$ vs $p_{open}$, respectively, and clearly illustrate the importance of sequence to the unwinding kinetics of XPD. The backstepping probability $p_-$ depended inversely on $p_{open}$. More strikingly, $k_F$ increased by a factor of four as $p_{open}$ increases, indicating that XPD relies heavily on thermal opening of downstream base pairs to unwind the hairpin. In contrast, the reverse rate $k_R$ displayed only a weak relation to $p_{open}$. Sub-saturating ATP concentration (*Figure 6D–I*) yielded weaker sequence dependences since $k_F$ was limited by binding of ATP, rather than unwinding the hairpin ('Materials and methods').

## Discussion

These base pair stepping dynamics reveal important aspects of XPD mechanism. In contrast to 3′–5′ SF1A helicases, which must remain in register with DNA because its bases are intercalated between aromatic residues of the active site (*Velankar et al., 1999*; *Lee and Yang, 2006*), SF1B and all SF2 helicases and related dsDNA translocases interact with the phosphodiester backbone (*Singleton et al., 2007*; *Fairman-Williams et al., 2010*; *Beyer et al., 2013*; *Raney et al., 2013*). Such interaction may allow for a non-unitary step size (*Moffitt et al., 2009*). Our high-resolution measurements of XPD steps establish unequivocally that this SF2B helicase unwinds DNA in uniform 1-bp steps, yet is an inefficient helicase. Unwinding is mitigated by frequent backsteps and duplex re-annealing. Although 1-bp backsteps are more frequent as ATP is decreased, the backstepping probability remains non-zero at saturating ATP (*Figure 4E*). This indicates that 1-bp backsteps occur mainly when the helicase waits for the next ATP to bind, but also occasionally while in the ATP-bound state. Single-bp backsteps likely represent helicase slippage where XPD remains partially engaged with the ssDNA. (We distinguish this behavior from 'backslides' [*Figure 1D*, inset], in which XPD moves backwards rapidly by many base pairs in one step, and which probably represent a more serious disengagement from the ssDNA.) The nature of the interaction between the motor core of XPD and DNA backbone is likely responsible for backstepping. This would be in a stark contrast with SF1A helicases, whose movement requires shifting in 1 nt register due to interaction with the bases. These essential features in the observed stepping dynamics are captured in the minimal kinetic model in *Figure 7A*.

Large ~5-bp backsteps likely involve a different mechanism. Based on structural data, we propose that these events represent rearrangements of the translocating DNA strand relative to a secondary binding site. Whereas typical SF2 helicases contain a bipartite nucleic acid binding site that accommodates up to 8 nt (or bp) of nucleic acid (*Singleton et al., 2007*; *Fairman-Williams et al., 2010*; *Beyer et al., 2013*), XPD and related 5′–3′, FeS-containing helicases feature additional contacts with the translocating strand beyond this canonical binding site. Biochemical and structural analyses (*Kuper and Kisker, 2012*; *Kuper et al., 2012*; *Pugh et al., 2012*) indicate that the translocating strand passes from the canonical binding site through the opening formed by the two auxiliary domains (ARCH and FeS) and is guided by the residues of HD1 and FeS domains, forming a secondary DNA binding site to the wedge structure on the back of the FeS domain that splits the duplex. As shown in *Figure 7—figure supplement 1*, this site accommodates ~5 nt. We propose that the observed 5-bp backsteps occur when the translocating strand dissociates from this secondary binding site, allowing the 5 nt normally occluded to re-anneal with the complementary strand. This renders the helicase incompetent for unwinding until the point of duplex separation reengages the wedge structure, necessitating

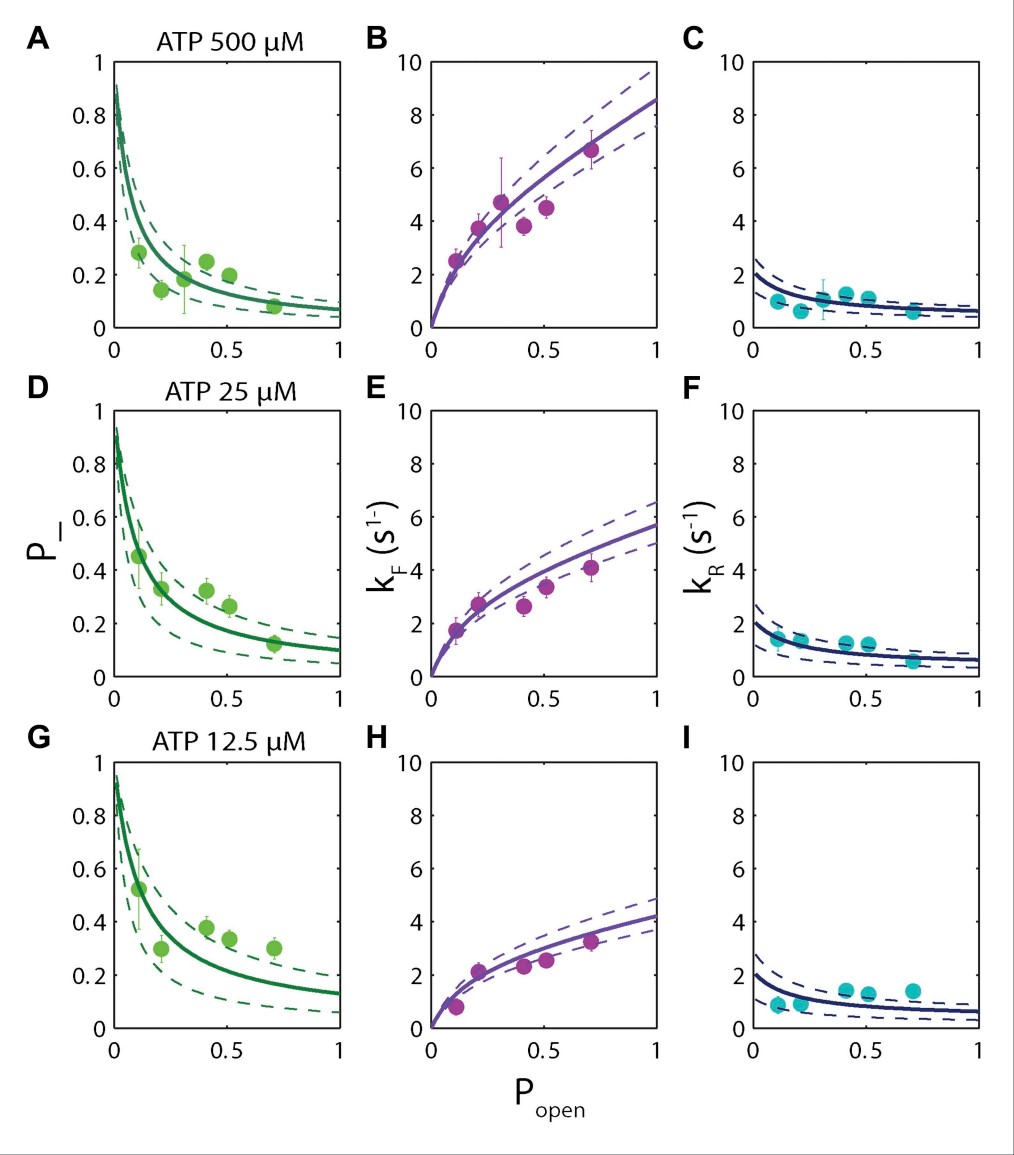

**Figure 6**. Dependence of kinetic parameters on DNA thermal opening probability. (**A**) Backstepping probability $p_-$, (**B**) Forward rate $k_F$, and (**C**) reverse rate $k_R$ vs $p_{open}$ at [ATP] = 500 µM. The solid lines represent the global fit to the kinetic model described in the text and 'Materials and methods'; dashed lines represent 95% confidence interval. The reduced $\chi^2$ value for the global fit to the data in panels (**B**), (**C**), (**E**), (**F**), (**H**) and (**I**) was 3.6. Error bars throughout denote SEM. (**D–F**) Same parameters at [ATP] = 25 µM. (**G–I**) Same parameters at [ATP] = 12.5 µM.

The following figure supplements are available for figure 6:

**Figure supplement 1**. 'Strictly passive' model fitting.

reopening of the 5 bp (**Pugh et al., 2012**). This is consistent with the observation that 5-bp backsteps are always followed by equal-size forward steps (**Figure 4D**). The observed force dependence of these events (**Figure 4—figure supplement 1**) may indicate that higher substrate tensions prevent the displaced strand from binding to the secondary site, stabilizing the translocating strand's interaction with this site. The 5-bp backstepping dynamics are included in the kinetic model of **Figure 7A**.

The fact that 5-bp backsteps were observed on a variety of sequences, including homo-AT templates (**Figure 5—figure supplement 3**, purple curve) disfavors an alternate mechanism in which the backsteps represent transient formation of a short hairpin on either strand. Other alternatives for 5-bp steps such as a spring-loaded mechanism (**Appleby et al., 2011**) are also implausible. Such a

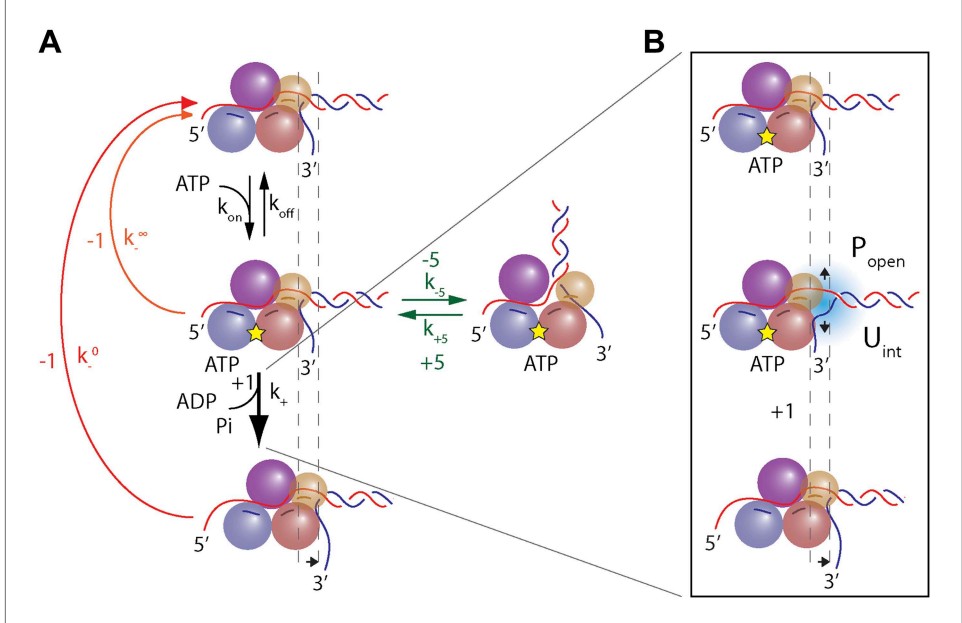

**Figure 7**. Model of XPD unwinding. (**A**) Minimal kinetic scheme for XPD stepping and backstepping. The main mechanochemical pathway consists of two rate-limiting steps: ATP binding and 1-bp unwinding (black arrows). 1-bp backsteps occur while the helicase is in the apo state (red arrow), or in the ATP-bound state (orange arrow). Large 5-bp backsteps involve rearrangement of the nucleo-protein complex (green arrows). The thickness of the arrows schematically represents the probability for each pathway. (**B**) Mechanism of duplex unwinding. Although XPD helicase has a destabilizing effect on the fork, unwinding depends strongly on spontaneous opening of the duplex. See main text and 'Materials and methods' for details.

The following figure supplements are available for figure 7:

**Figure supplement 1**. Structural model for regulatory secondary ssDNA binding site and large −5-bp backsteps.

**Figure supplement 2**. Representative traces and stochastic simulations of single XPD unwinding.

**Figure supplement 3**. Sequence dependence of simulated unwinding traces.

mechanism would require several hidden kinetic steps to occur per observed 5-bp step (i.e., 'loading of the spring'), resulting in a non-exponential dwell time distribution (*Myong et al., 2005*), inconsistent with our data (*Figure 4—figure supplement 2*). While we cannot rule out all alternative models and ternary complex structures will be required for further validation, the proposed mechanism provides the most reasonable explanation of our observations. The 5-bp back and forward steps illustrate the importance of the secondary binding site in controlling XPD helicase activity: it separates the motor core from the wedge structure that splits the duplex. Unwinding occurs only when the translocating strand resides within this secondary DNA binding site. This not only allows control of 5'–3' translocation and force generation by XPD (*Kuper et al., 2012*; *Pugh et al., 2012*), but also stalls the helicase at bulky UV-induced damage, thereby triggering nucleotide excision repair (*Mathieu et al., 2010*, *2013*).

The backstepping behavior of XPD is highly sequence-dependent (*Figures 5 and 6*). Confronted with stable sequences (i.e., when $p_{open}$ is small), the helicase exhibits a higher frequency of backsteps (*Figure 6A*). This occurs primarily because the rate of stepping forward $k_F$ is suppressed (*Figure 6B*). The linear dependence of $k_F$ on $p_{open}$ at saturating ATP suggests a 'passive' unwinding mechanism in which the helicase captures spontaneously opened base pairs at the ssDNA/dsDNA junction. However, a strictly passive model (*Betterton and Jülicher, 2005*; *Delagoutte and von Hippel, 2002*; *Johnson et al., 2007*; *Manosas et al., 2010*) requires that the rate be strictly proportional to $p_{open}$, whereas the observed linear trend exhibits an offset. This feature is indicative of an intrinsic destabilization of the duplex by the helicase, characteristic of an 'active' mechanism (see 'Materials and methods').

Based on the minimal kinetic scheme in *Figure 7A*, we fitted the kinetic data globally (dashed lines, *Figure 4E–F*; see 'Materials and methods'). To model quantitatively the dependence of the stepping rates on $p_{open}$ and any destabilizing interaction, we used the theoretical approach of *Betterton and Jülicher (2005)*. A global fit to the minimal kinetic scheme incorporating sequence dependence (solid lines, *Figure 6A–I*) suggests 'partially' active unwinding (*Figure 7B*). The fork is destabilized by 1.9 $k_BT$ through interaction with the protein, yet still slows the XPD's progress. (In contrast, an alternative 'strictly passive' unwinding model does not yield as good a fit to the data; *Figure 6—figure supplement 1*). The parameters of the best global fit are summarized in *Table 2*. We highlight the fact that the fit yields a translocation rate $k_{trans} \approx 10$ nt/s (obtained in the limit that $p_{open} = 1$), in excellent agreement with that determined from independent measurements of XPD translocation on ssDNA (*Honda et al., 2009*). The global model correctly and simultaneously reproduces the dependences of the 1-bp and 5-bp backstep probabilities, their respective dwell times, and processivity on ATP (*Figure 4E,F*) and sequence (*Figure 6A–I*).

Based on the parameters of the global model, we also performed stochastic simulations to generate individual unwinding bursts. *Figure 7—figure supplement 2*, which compares representative data traces to simulated traces, demonstrates that the simulations reproduce many of the observed features in the data: frequent backstepping, 5-bp backward and forward steps, and the sequence dependent processivity of XPD helicase (*Figure 7—figure supplement 3*). (Additional parameters in the model would be required to incorporate XPD backsliding and dissociation observed at the end of the unwinding bursts [*Figure 1D*]. Therefore, these simulations as currently constructed only reproduce a single unwinding burst).

The origin of the low processivity of XPD (*Figure 1D*) is the sequence dependence. When $k_F$ and $k_R$ are equal, the average unwinding rate is zero. As seen in *Figure 6B,C*, this condition occurs when $p_{open} \approx 0.1$, which matches the value of $p_{open}$ at the hairpin position beyond which most helicases stall, for the three hairpin sequences assayed (*Figure 5C* and *Figure 5—figure supplement 2*). Based on this argument, one would predict the processivity to decrease at lower tensions, as the equilibrium is shifted toward the closed hairpin conformation and $p_{open}$ is decreased. As shown in *Figure 5—figure supplement 5*, not only is this decrease observed but $p_{open} \approx 0.1$ predicts the stall position to within 1 bp.

The biological function of this inefficient unwinding likely reflects the cellular role of XPD helicase, which requires unwinding of short duplex regions. The sequence-dependence of XPD's unwinding activity we report here makes XPD suitable for its function in NER and transcription. Having a highly processive and vigorous helicase uncoupled from its respective macromolecular machinery (such as TFIIH complex in the case of XPD) would be undesirable for the cell. Protein partners may play an important role in facilitating and regulating XPD helicase unwinding activity (*Pugh et al., 2008b*), akin to our use of tension in assisting the protein. The conserved domain architecture of all SF2B helicases (including human FANCJ, RTEL and CHLR1; *White, 2009*) predicts that these helicases will share a common unwinding mechanism with XPD, despite a broad range of cellular functions. We thus speculate that distinct protein partners play a role in defining helicase role. It is known that XPD has functions outside of the TFIIH complex; by associating with different partners, XPD can play a role in chromosome segregation (*Ito et al., 2010*) and in the cell's defense against retroviral infection (*Yoder et al., 2006*). These molecular associations may target the SF2B helicase to a DNA structure requiring it activity and activate the SF2B helicase either for DNA unwinding or for remodeling of nucleoprotein complexes. The basic mechanistic features underlying duplex unwinding by these helicases are

**Table 2.** Summary of model parameters

| | |
|---|---|
| $k_{on}^{eff}$ (µM$^{-1}$ s$^{-1}$) * | 0.67 ± 0.09† |
| $k_+$ (s$^{-1}$) ‡ | 9.9 ± 1.0 |
| $k_*^*$ (s$^{-1}$) ‡ | 0.7 ± 0.3 |
| $k^0$ (s$^{-1}$) ‡ | 0.6 ± 0.5 |
| $k_{+5}$ (s$^{-1}$) ‡ | 6.8 ± 0.6 |
| $k_{-5}$ (s$^{-1}$) ‡ | 0.8 ± 0.2 |
| $f$ | 0.36 |
| $U_{int}$ ($k_BT$) | 1.9 |

*The global fit did not allow us to determine $k_{on}$ and $k_{off}$ individually. Instead, we estimated the 'effective' ATP binding rate constant $k_{on}^{eff} = k_{on} / \left(1 + \dfrac{k_{off}}{k_+}\right)$.

†Errors represent SD and were determined by bootstrapping.

‡These rate constants represent the forward and backward translocation rate constants in the limit that $p_{open} = 1$, that is, in the absence of a duplex to unwind.

likely to be conserved for all SF2B enzymes and need to be understood before studies of the effect of partner proteins can be carried out. We anticipate that our high-resolution assay may help decipher the mechanisms and regulation of these helicases in the future.

## Materials and methods

### Optical tweezers

High-resolution dual-trap optical tweezers based on a previously reported design (*Bustamante et al., 2008*) were used to study XPD helicase unwinding. The optical traps were calibrated using standard procedures (*Berg-Sorensen and Flyvbjerg, 2004*; *Bustamante et al., 2008*). In all measurements both traps had a stiffness $k = 0.3$ pN/nm. All data were acquired using custom LabVIEW software (8.2; National Instruments, Austin, TX). The base pair-resolution stepping traces were collected using force feedback (*Neuman and Block, 2004*) at a 1-kHz sampling rate, and boxcar filtered to a lower frequency as indicated in the text.

### DNA hairpin construct

The hairpin construct was adapted from a previous protocol (*Woodside et al., 2006*), and consisted of three separate fragments ligated together after synthesis and purification (*Figure 1—figure supplement 1*): 'Right handle' (R), 'Hairpin' (H), 'Left Handle' (L). R was synthesized from a 1.5-kb PCR-amplified section of the pBR322 plasmid (New England Biolabs, Ipswich, MA) using a 5'-digoxigenin-modified forward primer and a reverse primer containing one abasic site. The abasic site left a 29-nt 5' overhang upon PCR amplification ("Auto-sticky PCR"; *Gal et al., 1999*) that annealed to a complementary sequence in the 89-bp hairpin H. L was PCR-amplified from the same plasmid using a 5'-biotin-modified primer and cut to 1550-bp length with the PspGI restriction enzyme (New England Biolabs, Ipswich, MA), leaving a 5-nt 5' overhang. H was composed of a long oligonucleotide containing (from 5' to 3') the complementary sequence to the overhang in L, a $(dT)_n$ 'loading site' for helicase binding ($n = 10$ nt unless otherwise noted), and a 153-nt sequence containing the hairpin and a $(dT)_4$ tetraloop. The construct allowed for a hairpin with different length or sequence to be easily substituted. Three hairpin constructs were synthesized: (1) 'sequence 1' containing a random 49% GC sequence: GGC TGA TAG CTG AGC GGT CGG TAT TTC AAA AGT CAA CGT ACT GAT CAC GCT GGA TCC TAG AGT CAA CGT ACT GAT CAC GCT GGA TCC TA; (2) 'sequence 2' which consisted of 'sequence 1' with all GC pairs replaced with AT and vice versa: AAT CAG CGA TCA GAT AAC TAA CGC CCT GGG GAC TGG TAC GTC AGC TGT ATC AAG CTT CGA GAC TGG TAC GTC AGC TGT ATC AAG CTT CG; (3) 'sequence 3' which consisted of alternating ~30-bp homo-AT, homo-GC, homo-AT sequences: TTA ATA AAT AAA TAA ATA AAA TAA ATA AAG GGC GGC GGG CGG GCG GGC GGG CGG GCG GAT TAA TAA ATA AAT AAA TAA AAT AAA TAA AA. All oligonucleotides were from Integrated DNA Technologies (Coralville, IA).

### Laminar flow cell

Laminar flow cells (*Brewer and Bianco, 2008*; *Figure 1—figure supplement 3*) consisted of two microscope cover glasses (60 × 24 × 0.013 mm; Fisher Scientific, Pittsburgh, PA) sandwiching a piece of melted parafilm (Nescofilm; Karlan, Phoenix, AZ). Eight small holes (four on each side) were cut on the top cover glass by a laser engraver system (VLS2.30; Universal Laser Systems, Scottsdale, AZ), to which four inlet and outlet tubes were connected. Three channels were engraved into the parafilm. The top and bottom channels (yellow and green, *Figure 1—figure supplement 3*) were used to flow in streptavidin and anti-DIG coated beads, respectively. Glass capillaries (OD = 100 ± 10 μm, ID = 25.0 ± 6.4 μm; Garner Glass Co., Claremont, CA) through which beads could flow connected the top and bottom channels to the central channel, where the optical trap measurements were performed. In the central channel, separate streams from two inlets (blue and red) merged at the tip of a 'parafilm triangle'. Since the flow was laminar, a sharp interface between streams was maintained. A syringe pump (PHD 2000 Infusion; Harvard Apparatus, Holliston, MA) was used to control the flow of different buffers into the cell, at a rate of 100 μl/hr ($v = 140$ μm/s linear flow speed). A trapped molecule could be displaced across the laminar flow interface by moving the flow cell relative to the traps with a motorized linear stage (Model ESP 300; Universal Motion Controller, Newport, Bozeman, MT). We estimate that the interface width, at a typical location in the cell and for a small molecule like ATP, was <0.3 mm (defined as the distance from 10% to 90% maximum concentration). With a typical stage speed of 0.2 mm/s, solution exchange occurred within ~2 s.

## Single-molecule Helicase Experiment protocol

*F. acidarmanus* XPD was purified as previously described (*Pugh et al., 2008a*, *2008b*). The working buffer consisted of 100 mM Tris–HCl (pH 7.6), 20 mM NaCl, 2 mM DTT, 3 mM MgCl$_2$, 0.1 mg/mL BSA and oxygen scavenging system (0.5 mg/mL glucose oxidase [Sigma-Aldrich, St. Louis, MO], 0.1 mg/mL catalase [Sigma-Aldrich, St. Louis, MO], and 0.4% glucose) to increase tether lifetime and reduce photodamage (*Landry et al., 2009*); to this buffer, varying concentrations of ATP (Sigma-Aldrich, St. Louis, MO) and XPD were added. Helicase activity was detected from unwinding of a DNA hairpin. The hairpin constructs were functionalized with 5′ biotin and 5′ digoxigenin ends, and were tethered between a trapped streptavidin-coated bead (0.79 μm; Spherotech Inc., Lake Forest, IL) and an anti-digoxigenin-coated protein G bead (0.86 μm; Spherotech Inc., Lake Forest, IL) in the laminar flow cell.

In a typical experiment, the two streams merging in the central channel contained buffer with ATP (0–500 μM, blue stream; *Figure 1C* and *Figure 1—figure supplement 3*), and buffer with XPD helicase (6 nM, red stream). A streptavidin and anti-DIG bead were captured by the traps, and a single tether formed in the ATP stream. A force-extension curve (FEC) was taken for every tether formed to verify proper behavior. The tether was moved into the XPD stream at low tension (<2 pN). Once inside the XPD stream (position 1, *Figure 1C*), the tension was increased to a fixed value below the hairpin unfolding force and maintained using force feedback, and data acquisition started (*Figure 1D*). After a short incubation period to allow XPD to bind, the tether was moved to the ATP stream (position 3, *Figure 1C*). A different protocol was used for the XPD titration experiments in *Figure 2F,H*. In this case, the two merging streams contained blank buffer (green stream) and buffer mixed with both XPD and 500 μM ATP ([XPD] = 0.2–60 nM, red stream). A single tether was formed at position (1), and then moved along the dashed-line path across the stream interface (2) and into the XPD + ATP-rich stream (3).

## Step size analysis

Two step analysis methods were used: the pairwise distance distribution (PWD) (*Abbondanzieri et al., 2005*; *Dumont et al., 2006*; *Moffitt et al., 2009*) and the step-fitting algorithm developed by *Kerssemakers et al. (2006)*. In the first method, unwinding segments from bursts were selected and boxcar filtered to 25 Hz. The PWD from a selection of the best traces (*Table 1*) were averaged together for each ATP concentration (*Figure 3B*). In the second method, the unwinding traces were filtered and decimated to 250 Hz, and run through the step-fitting algorithm. Dwell times <20 ms and step sizes <0.4 bp (corresponding to the noise in extension; *Figure 1E*) were removed. Backstep probabilities were calculated throughout using the Laplace estimator ($n_{success}$ + 1)/($N_{trial}$ + 2). All data analysis was performed on custom Matlab software (R2010a; MathWorks, Inc., Natick, MA).

## Hairpin force-extension curve fitting

The hairpin elastic behavior was modeled using the Worm-like Chain model (WLC) (*Bustamante et al., 1994*). The parameters used for dsDNA were as follows: the persistence length was $P_{ds}$ = 50 nm, stretch modulus $S_{ds}$ = 1000 pN, and the contour length per base pair $h_{ds}$ = 0.34 nm/bp. For ssDNA, the parameters used were $P_{ss}$ = 1.0 nm, $S_{ss}$ = 1000 pN, $h_{ss}$ = 0.6 nm/nt, consistent with previous values (*Murphy et al., 2004*; *Dumont et al., 2006*; *Woodside et al., 2006*). These parameters were determined from fits of FEC of two 'test' molecules, a 3.4-kb dsDNA construct and a 3.25-kb ds-ssDNA hybrid construct consisting of 1.55-kb and 1.7-kbp dsDNA handles ligated to a central 70-dT ssDNA segment. The FEC were obtained under the same buffer conditions as our XPD helicase measurements.

To fit the unfolding transition in our hairpins, we utilized the approach of *Huguet et al. (2010)*. The parameters of the calculation were: the calibrated trap stiffness, the WLC parameters for dsDNA and ssDNA and the hairpin base pairing energies. These were obtained from the measured 10 nearest-neighbor (*Borer et al., 1974*; *Santalucia, 1998*) and 1 loop free energies, and from correction factors $\left[ Mon^+ \right] = \left[ Tris^+ \right] + \left[ Na^+ \right] + \beta \sqrt{\left[ Mg^{2+} \right]}$ to account for the effect of monovalent ions (*Huguet et al., 2010*). To account for divalent ions in our buffers, we used the empirical formula (*Owczarzy et al., 2008*) to estimate an 'effective' ionic concentration. We allowed $\beta$ to be a fitting parameter, with $\beta \sim 8$ achieving the best global fit to the FECs for all three hairpin sequences (*Figure 1—figure supplement 2*).

Based on these base pairing free energies, we calculated the probability, $p_{open}(n,F)$, that the hairpin fork opens by one or more base pairs downstream of each fork position $n$ along the hairpin sequence

given the force $F$. We followed the same approach as *Johnson et al. (2007)* calculating the free energy of opening base pairs 1 to $n$ from $\Delta G_{tot}(n) = \sum_{i=n+1}^{l_{ss}} \Delta G_{bp}(i) - 2n \int_0^F x(F')\,dF'$, the sum of base pairing energies for base pairs $n + 1$ to $l_{ss}$ and the energy of stretching $2n$ released nucleotides at force $F$ ($x(F)$ is the extension of one ssDNA nucleotide). All modeling was performed on custom Matlab software (R2010a, MathWorks, Inc.).

While AT% is used to describe DNA sequences typically, $p_{open}$ is a preferable parameter for several reasons. First, AT% must be calculated over an arbitrarily chosen window at each hairpin position, yet AT% values depend on window size. In contrast, $p_{open}$ quantifies the probability that one or more base pairs downstream of the ss-dsDNA junction open spontaneously due to thermal fluctuations and thus depends on the sequence of 'all' downstream base pairs, with appropriate statistical weights. *Figure 5— figure supplement 1* shows that, for all hairpin sequences used in this study, $p_{open}$ correlates well with the AT% (over the appropriately sized window). Secondly, since $p_{open}$ quantifies thermal fraying of the duplex, it assesses how 'active' or 'passive' a helicase is better than AT%. Finally, unlike AT%, $p_{open}$ can be computed and compared directly to models of helicase mechanism (*Betterton and Jülicher, 2005*).

## Modeling the kinetics of XPD unwinding

We devised a simple, minimal kinetic model that can quantitatively describe the unwinding data. Several essential features in the data must be captured: the model must provide a mechanism by which (i) the mean dwell time satisfies Michaelis-Menten-like kinetics (*Figure 4F*), (ii) the backstepping probability $p_-$ increases as ATP is decreased, yet remains nonzero at saturating ATP (*Figure 4E*), (iii) large (~5-bp) backsteps are always followed by large forward steps (*Figure 4D*), and (iv) the probability of taking large backsteps exhibits a weak increase with increasing ATP concentration (*Figure 4E*). All of these criteria are satisfied by the kinetic scheme depicted in *Figure 7*.

Criterion (i) requires that the mechanochemical cycle of the helicase contains a minimum of two kinetic steps: an ATP binding step, which becomes rate-limiting at low ATP concentrations, followed by an unwinding step, rate-limiting at saturating ATP concentrations. The rate constants for these steps are $k_{on}$ (for the sake of generality we assume binding can be reversible, with a dissociation rate constant $k_{off}$) and $k_+$, respectively. Criterion (ii) requires that there exist two competing pathways for backstepping. In one, the motor can backstep from the nucleotide-free state with rate $k_-^0$. As ATP concentration is decreased, the motor resides longer in this state, thus increasing the probability of backstepping. In the second pathway, the ATP-bound motor may backstep with rate $k_-^\infty$. This provides a mechanism by which the motor can backstep even as ATP concentration becomes saturating. Criterion (iii) requires that the large 5-bp backsteps take the motor to an off-pathway state, in which return to the main mechanochemical cycle can only occur through a forward 5-bp step. The dependence on ATP from criterion (iv) is ensured by making entry into this off-pathway state occur from the nucleotide-bound state.

Based on this scheme, we determined several relevant kinetic parameters to compare to our measurements. In terms of the mean dwell time and backstepping probability $p_-$, the average unwinding velocity is given by

$$v = d\,\frac{p_+ - p_-}{\tau},$$

where $d = 1$ bp is the step size and $p_+ = 1 - p_-$ is the forward stepping probability. We defined forward and reverse rate constants $k_F$ and $k_R$ (*Norstrom et al., 2010*) such that

$$v = d\left(k_F - k_R\right),$$

from which it follows that

$$p_- = \frac{k_R}{k_R + k_F} \text{ and } \tau = \frac{1}{k_R + k_F}.$$

Using a general approach for solving kinetic models (**Chemla et al., 2008**), we calculated the following kinetic parameters based on the proposed scheme:

$$p_- = \frac{\dfrac{k_-^\infty}{k_+}[ATP] + \dfrac{k_-^0}{k_{on}}\left(1 + \dfrac{k_{off}}{k_+} + \dfrac{k_-^\infty}{k_+}\right)}{[ATP]\left(1 + \dfrac{k_-^\infty}{k_+}\right) + \dfrac{k_-^0}{k_{on}}\left(1 + \dfrac{k_{off}}{k_+} + \dfrac{k_-^\infty}{k_+}\right)}, \tag{1}$$

$$\tau = \frac{\dfrac{1}{k_+}\left(1 + \dfrac{k_{-5}}{k_{+5}}\right)[ATP] + \dfrac{1}{k_{on}}\left(1 + \dfrac{k_{off}}{k_+} + \dfrac{k_-^\infty}{k_+}\right)}{[ATP]\left(1 + \dfrac{k_-^\infty}{k_+}\right) + \dfrac{k_-^0}{k_{on}}\left(1 + \dfrac{k_{off}}{k_+} + \dfrac{k_-^\infty}{k_+}\right)}, \tag{2}$$

$$k_F = \frac{[ATP]}{\dfrac{1}{k_+}\left(1 + \dfrac{k_{-5}}{k_{+5}}\right)[ATP] + \dfrac{1}{k_{on}}\left(1 + \dfrac{k_{off}}{k_+} + \dfrac{k_-^\infty}{k_+}\right)}, \tag{3}$$

and

$$k_R = \frac{\dfrac{k_-^\infty}{k_+}[ATP] + \dfrac{k_-^0}{k_{on}}\left(1 + \dfrac{k_{off}}{k_+} + \dfrac{k_-^\infty}{k_+}\right)}{\dfrac{1}{k_+}\left(1 + \dfrac{k_{-5}}{k_{+5}}\right)[ATP] + \dfrac{1}{k_{on}}\left(1 + \dfrac{k_{off}}{k_+} + \dfrac{k_-^\infty}{k_+}\right)}. \tag{4}$$

Several limits in **Equations 1–4** are illuminating. For simplicity we consider that there are no 5-bp backsteps, $k_{-5} = 0$. At saturating ATP, $k_F = k_+$, the forward stepping rate, and $k_R = k_-^\infty$, the backstepping rate, and the backstepping probability is given by the kinetic competition between the two. When backstepping from the nucleotide-free state is removed ($k_-^0 = 0$), $p_-$ is constant, independent of ATP. When backstepping from the ATP-bound state is removed ($k_-^\infty = 0$), $p_-$ depends inversely on ATP and approaches zero as ATP becomes large. When both backstepping rates $k_-^\infty$ and $k_-^0$ are set to zero, the dwell time $\tau = 1/k_F$ reduces to a sum of the inverses of the forward stepping rate constant $k_+$ and an 'effective' ATP binding rate constant $k_{on}^{eff} = k_{on}[ATP]/\left(1 + \dfrac{k_{off}}{k_+}\right)$. The same approach can also be used to determine the large backstep probability:

$$p_{-5} = \frac{\dfrac{k_{-5}}{k_+}\left([ATP] + \dfrac{k_-^0}{k_{on}}\right)}{[ATP]\left(1 + \dfrac{k_-^\infty}{k_+} + \dfrac{k_{-5}}{k_+}\right) + \dfrac{k_-^0}{k_{on}}\left(1 + \dfrac{k_{off}}{k_+} + \dfrac{k_-^\infty}{k_+} + \dfrac{k_{-5}}{k_+}\right)}. \tag{5}$$

## Modeling the sequence dependence of XPD unwinding

In **Equations 1–4**, the dependence on DNA sequence has not been made explicit. In general, we expect the forward stepping rate $k_+$ and potentially both backstepping rates $k_-^\infty$ and $k_-^0$ to depend on the energy of hairpin opening and the interaction between the helicase and hairpin fork. (In contrast, we do not expect purely chemical steps such as ATP binding and dissociation to display such a dependence). To model this effect, we used the theoretical approach developed by **Betterton and Jülicher (2005)**. There are many ways to formulate an interaction between helicase and fork. In the simplest implementation (called the "one-step potential"), the helicase destabilizes the base pair at the hairpin fork by the interaction energy $U_{int}$. A second parameter, $f$, which ranges from 0 to 1, determines whether this interaction accelerates the rate of hairpin opening (the limit $f{\to}0$), decreases that of hairpin closing ($f{\to}1$), or both ($0 < f < 1$).

In the limit that the hairpin opening and closing rates are much faster than the rates of helicase stepping or backstepping (which we expect to hold to a very good approximation; **Betterton and**

*Jülicher, 2005*; *Manosas et al., 2010*), one can show that the forward and backward stepping rates are given by

$$k_{\pm} = g_{\pm}\left(p_{open}, f, U_{int}\right) k_{\pm}^{trans},$$  (6)

where $k_{\pm}^{trans}$ are the forward and backward rates for translocation on ssDNA (i.e., the rates in the absence of a duplex to unwind), respectively. The factors $g_{\pm}$ quantify the effect of the helicase-fork interaction and the native duplex stability on the stepping rate; $p_{open}$ is the probability the fork opens given the base pairing energy and the destabilizing effect of tension (*Johnson et al., 2007*; *Huguet et al., 2010*). *Betterton and Jülicher (2005)* show that, in this simple interaction model, these factors are given by

$$g_{+} = \frac{p_{open}\left(1 - e^{-fU_{int}/k_BT}\right) + e^{-fU_{int}/k_BT}}{p_{open}\left(1 - e^{-U_{int}/k_BT}\right) + e^{-U_{int}/k_BT}} p_{open}$$  (7)

$$g_{-} = \frac{p_{open}\left(1 - e^{-fU_{int}/k_BT}\right) + e^{-fU_{int}/k_BT}}{p_{open}\left(1 - e^{-U_{int}/k_BT}\right) + e^{-U_{int}/k_BT}},$$  (8)

where $k_B$ is the Boltzmann constant, and $T$ is the absolute temperature.

Several limits in *Equations 6–8* are illuminating. In the limit that $U_{int} = 0$, $g_{+} = p_{open}$. The helicase can only step forward when the hairpin fork spontaneously opens by thermal fluctuation; as a result, unwinding is considered 'strictly passive' (*Betterton and Jülicher, 2005*; *Delagoutte and von Hippel, 2002*; *Johnson et al., 2007*; *Manosas et al., 2010*). In the limit that $f = 0$ and $U_{int}$ is very large (>>1 $k_BT$), $g_{+} = 1$. The helicase accelerates the rate of fork opening to such an extent that unwinding is limited only by its translocation rate $k_{+}^{trans}$ on ssDNA; here unwinding is referred to as 'optimally active' (*Johnson et al., 2007*). The intermediate regime is most relevant to our work. When $f > 0$ and $U_{int}$ is reasonably large, the denominators in $g_{\pm}$ are approximately equal to $p_{open}$, and $g_{+}$ takes the form $g_{+} \approx x + (1 - x)p_{open}$, valid for $p_{open} > 1/\left(e^{U_{int}/k_BT} + 1\right)$. Thus, the forward stepping rate varies linearly with $p_{open}$, but the linear trend intercepts at a nonzero value of $x \approx e^{-fU_{int}/k_BT}$ as $p_{open}$ is extrapolated to zero. This is the behavior observed in *Figure 6B*, plotting $k_{+}$ against $p_{open}$. The non-zero intercept is a manifestation of the interaction between helicase and hairpin fork.

## Fitting XPD unwinding data to kinetic model

A complete kinetic model combining *Equations 1–8* was used to perform a global fit of the kinetic data obtained on XPD helicase (*Figures 4E,F and 6A–I*). The four quantities plotted as a function of ATP in *Figure 4E,F*—the probabilities for 1-bp and 5-bp backsteps $p_{-}$ and $p_{-5}$, the dwell times for 1-bp steps and 5-bp backsteps $\tau$ and $\tau_5$—averaged over the effect of hairpin sequence. Thus, we fitted these data to expressions based on *Equations 1*, *2*, and *5*, averaging over the range of $p_{open}$ accessed in the measurements. For instance, in *Figure 4E* the backstepping probability was fit to

$$\langle p_{-} \rangle = \sum p_{-}\left(p_{open}\right) \rho\left(p_{open}\right),$$

where $\rho(p_{open})$ is the exact distribution of $p_{open}$ obtained in our measurements. Similar expressions were used for the other kinetic parameters.

More specifically, we used expressions of the form

$$p_{-} = \frac{B[ATP] + A(1 + B + C)}{[ATP](1 + B) + A(1 + B + C)},$$  (9)

$$p_{-5} = \frac{D([ATP] + A)}{[ATP](1 + B + D) + A(1 + B + C + D)},$$  (10)

$$\tau = \frac{(b + D\tau_5)[ATP] + a(1 + B + C)}{[ATP](1 + B) + A(1 + B + C)}, \text{and}$$  (11)

$$\tau_5 = \frac{1}{k_{+5}}, \qquad (12)$$

based on *Equations 1*, *2*, and *5* for the four quantities of interest. Six fitting parameters were defined as: $a \equiv 1/k_{on}$, $b \equiv 1/k_+$, $A \equiv k_-^0/k_{on}$, $B \equiv k_-^\infty/k_+$, $C \equiv k_{off}/k_+$, $D \equiv k_{-5}/k_+$. Those parameters dependent on sequence were functions of $g_+$ and $g_-$ according to *Equations 6–8*.

For a given model of the sequence dependence parametrized by $f$ and $U_{int}$, we fitted all four data sets in *Figure 4E,F* to the $p_{open}$-averaged *Equations 9–12* and determined the parameters $a$, $b$, $A$, $B$, $C$, and $D$. $\chi^2$-minimization was used to determine the best global fit. Reduced $\chi^2$ values (*Bevington and Robinson, 2002*) are provided in the figure caption. From $a$, $b$, $A$, $B$, $C$, and $D$ we could determine all individual rate constants. We found that the fits depended only weakly on $C$ and did not allow us to determine $C$ and $a$ individually with any accuracy. Instead, fits could determine the effective ATP binding rate constant $k_{on}^{eff} = k_{on}[ATP]/(1 + k_{off}/k_+)$ defined above accurately.

We determined the optimal values of the two sequence dependence parameters $f$ and $U_{int}$ by fitting the forward and reverse rates $k_F$ and $k_R$ vs $p_{open}$ in *Figure 6B,C,E,F,H,I*, minimizing $\chi^2$ to determine the best global fit to all six data sets. The rate constants listed in *Table 2* were obtained from the values of $a$, $b$, $A$, $B$, $C$, and $D$ at the best-fit values of $f$ and $U_{int}$. Fits in *Figures 4E,F and 6A–I* correspond to those parameter values. An alternate 'strictly passive model' fit with $U_{int} = 0 = f$ is shown in *Figure 6—figure supplement 1*. Reduced $\chi^2$ values for both global fits are compared in the figure caption. A purely passive model does not fit the data as well as the partially active model presented in the text.

The eight parameters used to generate the fits in *Figures 4E,F and 6A–I*—six kinetic rate constants and two sequence dependence parameters—are listed in *Table 2*. Despite the apparent high number of parameters, the data determine these precisely. Four independent quantities measured as a function of ATP are plotted in *Figure 4E,F*: $p_-$, $p_{-5}$, $\tau$ and $\tau_5$. If these quantities had been fitted individually, a minimum of seven independent kinetic parameters would have been required—two for $p_-$, $p_{-5}$, and $\tau$ each, which depend on ATP, and one $\tau_5$—a higher number than that used in our model. Similarly, six independent plots of $k_F$ and $k_R$ vs $p_{open}$ are shown in *Figure 6B,C,E,F,H,I*. If these had been fitted independently, a number >2 of parameters would have been required to capture the dependence on $p_{open}$. In our global fit, all six plots were simultaneously fitted with only two parameters, $f$ and $U_{int}$, from the model of *Betterton and Jülicher (2005)*.

## Stochastic simulations of XPD unwinding

In order to test the validity of our model, we performed stochastic simulations of XPD unwinding using custom Matlab programs. The simulations followed closely a master equation formalism and used the kinetic model described above. Using as input the number of states, transition rate constants connecting these states, step sizes for forward and backward 1-bp and 5-bp steps (given by *Figure 7* and *Table 2*), and the hairpin sequences (see 'Materials and methods'), the simulations generated time courses of the states occupied by the motor and the DNA length unwound as a function of time. Representative simulated unwinding bursts are shown in *Figure 7—figure supplement 2*.

## Acknowledgements

We thank all the members of the Chemla and Spies laboratories for their generous advice.

## Additional information

### Funding

| Funder | Grant reference number | Author |
|---|---|---|
| National Science Foundation | MCB 09-52442 | Yann R Chemla |
| National Institutes of Health | R21 RR025341 A | Yann R Chemla |
| Howard Hughes Medical Institute | | Maria Spies |
| Burroughs-Wellcome Fund | | Yann R Chemla |

The funders had no role in study design, data collection and interpretation, or the decision to submit the work for publication.

## Author contributions

ZQ, Conception and design, Acquisition of data, Analysis and interpretation of data, Drafting or revising the article; RAP, Drafting or revising the article, Contributed unpublished essential data or reagents; MS, Conception and design, Drafting or revising the article, Contributed unpublished essential data or reagents; YRC, Conception and design, Analysis and interpretation of data, Drafting or revising the article

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
