## [Decision Letter]

Thank you for choosing to send your work entitled “Sequence-Dependent Base-Pair Stepping Dynamics in XPD Helicase Unwinding” for consideration at *eLife*. Your article has been evaluated by a Senior editor and 3 reviewers, one of whom, Leemor Joshua-Tor, is a member of our Board of Reviewing Editors.

The Reviewing editor and the other reviewers discussed their comments before we reached this decision, and the Reviewing editor has assembled the following comments based on the reviewers' reports.

The reviewers were impressed with the technical achievements described. An accurate step size has been determined directly for only a handful of motors, and so the high-resolution measurements described in this paper for this important helicase are impressive. The evidence for single base-step forward unwinding is convincing.

1) The reviewers do, however, have some concerns about the complex behavior consisting of the other step sizes: –1, and especially the +/–5bp, which are unique to this helicase. The suggestion that this is due to DNA release from a secondary site on the helicase has not been substantiated, nor has a satisfying biological explanation been provided. Figure 4 does not clearly show that the probability of taking 5bp steps depends on ATP concentration, as required by the modeling. In addition, it is interesting that the 5bp steps is smaller than the minimal nucleic acid binding site of this class of helicases, which is 6–7 bases. Perhaps a spring-loaded mechanism, such as that suggested by Appleby et al (JMB 2011), should be considered.

2) Of most concern with regard to modeling the complex stepping behavior is that the model contains eight fitting parameters. Such a large number of parameters could fit almost any small data set. In addition, the analyzed data in Figure 6 are rather scattered and many fits appear to be poor. What are the reduced chi-square values for these fits? More stringent tests for this model are called for. For example, can the model predict kinetic traces such as those in Figure 1, which shows five rounds of repetitive unwinding bursts?

---

## [Author Response]

*1) The reviewers do, however, have some concerns about the complex behavior consisting of the other step sizes: –1, and especially the +/–5bp, which are unique to this helicase. The suggestion that this is due to DNA release from a secondary site on the helicase has not been substantiated, nor has a satisfying biological explanation been provided.*
Figure 4
*does not clearly show that the probability of taking 5bp steps depends on ATP concentration, as required by the modeling. In addition, it is interesting that the 5bp steps is smaller than the minimal nucleic acid binding site of this class of helicases, which is 6–7 bases. Perhaps a spring-loaded mechanism, such as that suggested by Appleby et al (JMB 2011), should be considered*.

As correctly stated by the reviewers, a typical SF2 helicase contains a bipartite nucleic acid binding site that accommodates up to 8 nucleotides (or base pairs) of DNA or RNA. In addition to this canonical nucleic acid binding site, XPD and related FeS-containing helicases feature additional contacts with the translocating strand. Biochemical and structural analyses previously published by us and others (37; 59) indicate that the translocating strand passes from the canonical binding site through the opening formed by the two auxiliary domains (Arch and FeS), and is guided by the residues of HD1 and FeS domains forming a secondary DNA binding site to the wedge structure on the back of the FeS domain. This secondary DNA binding site should accommodate approximately 5 nt of the translocating strand. This site is important for (i) enabling and controlling 5’-3’ translocation and force generation by XPD, (ii) for positioning the ss-dsDNA junction near the strand separating wedge, and (iii) for damage verification by XPD (Mathieu et al. *Curr. Biol.* 2013). If the translocating strand were to dissociate from the secondary binding site, the 5 nt occluded by this site would re-anneal with the complementary strand, resulting in a 5-bp back step, as observed in our data. To re-initiate unwinding, the point of duplex separation has to reengage the wedge structure necessitating reopening of the 5 bp. In our data, a 5-bp forward step is always observed following a 5-bp backward step and is never detected on its own, consistent with this structural model.

The 5-bp back and forward steps illustrate the importance of the secondary binding site in controlling XPD helicase activity: it separates the motor core from the wedge structure that splits the duplex. Unwinding occurs only when the translocating strand resides within this secondary DNA binding site. This not only allows control of the unwinding rate (37; 59), but also stalls the helicase at bulky UV-induced damage site, thereby triggering nucleotide excision repair (Mathieu et al. *PNAS*, 2010; Mathieu et al. *Curr. Biol.* 2013).

While we cannot rule out all alternative models and ternary complex structures will be required for further validation, the proposed mechanism provides the most reasonable explanation of our observations. In contrast, the spring-loaded mechanism suggested by the reviewers appears far less plausible. In this mechanism, one would expect multiple unwinding steps reflecting “loading of the spring” accompanied by sequestering of the unwound DNA, followed by release of this DNA, which would be observed as a large forward step. It is difficult to imagine how such a model could explain how 5-bp backsteps are always followed by 5-bp forward steps, while also accounting for 1-bp forward and backward steps. Moreover, a spring-loaded mechanism would require several hidden kinetic steps to occur per observed step, resulting in a non-exponential dwell time distribution (see, for example, Myong et al. *Nature* 2005). This is inconsistent with our data; the dwell times for forward 5-bp steps are exponentially distributed, indicative of a single kinetic event.

In our revised manuscript, we have provided a more extensive discussion of these points. In addition, we have included a new Figure 4—figure supplement 2 showing that the dwell times for –5-bp backsteps are exponentially distributed.

*2) Of most concern with regard to modeling the complex stepping behavior is that the model contains eight fitting parameters. Such a large number of parameters could fit almost any small data set. In addition, the analyzed data in*
Figure 6
*are rather scattered and many fits appear to be poor. What are the reduced chi-square values for these fits? More stringent tests for this model are called for. For example, can the model predict kinetic traces such as those in*
Figure 1*, which shows five rounds of repetitive unwinding bursts*?

Our study design allows a wealth of information to be extracted from a relatively “small” dataset because individual steps are resolved. Each step contains information on directionality, step size, dwell time, and sequence. When multiplied by the several thousand steps detected (see Table 1) over varying ATP concentrations, the dataset can in fact provide strict constraints on a multi-parameter model.

Despite the apparent high number of fitting parameters, it is important to realize that these were in fact precisely determined by our data. To illustrate this point consider that our measurements yielded four independent quantities measured as a function of ATP (plotted in Figure 4): the probabilities for (i) 1-bp and (ii) 5-bp backsteps, the dwell times for (iii) 1-bp steps, and (iv) 5-bp backsteps. If these quantities had been fitted individually, a minimum of *seven* independent kinetic parameters would have been required to capture their dependence on ATP—two for each quantity, minus one for the 5-bp backstep dwell time that did not depend on ATP—rather than the *six* used in our model (see Table 2). (For instance, the dwell time for 1-bp steps would be fit to a Michaelis-Menten equation, yielding two parameters, *K*_*M*_ and *k*_*cat*_, etc) Similarly, the forward and backward rates (and the related backstepping probability, plotted in Figure 6) were measured as a function of sequence. According to the model of Betterton and Julicher (*Phys. Rev. E*., 2005), a minimum of *two* additional parameters, *f* and *U*_*int*_ (see Table 2), are required to capture the dependence on *P*_*open*_.

Rather than fit each quantity independently with a larger total set of parameters, we felt it was more instructive to propose a minimal model that would *globally* fit all datasets simultaneously. The global model parameters are in fact more strictly constrained than those of independent fits to the same set of plots.

In our revised manuscript, we have expanded a discussion of our fitting strategy in the Materials and methods. In addition, we have provided chi-squared values for all global fits in the figure captions. We also provide a fit to an alternate strictly passive unwinding model (i.e. with *U*_*int*_ = 0) with smaller numbers of parameters to show that the current model provides a more optimal fit (Figure 6—figure supplement 1).

The reviewers also raised a specific point about the ATP dependence of the 5-bp backstep probability (Figure 4). We agree that the dependence is quite weak, as mentioned in the manuscript text. One of the parameters in the global fit (C≡koff/k+) produces this ATP dependence with non-zero value. The chi-squared value is minimized for *C* > 0 although its dependence on *C* is weak (chi-squared values changed from 3.5 to 3.6 when varying *C* over a finite range), and the fits do not constrain the value of *C* very much. As described in the text (and see Table 2) we lump this parameter into an effective binding rate constantkoneff=kon/(1+koff/k+), which is more accurately constrained by the fits. The difference in the parameter values listed in Table 2 with *C* > 0 or with *C* = 0 is minimal, and this parameter is not essential to our model.

We have re-fitted our data to reflect changes to the data plots. The new fits are plotted in Figures 4 and 6, and the fit parameter values (which only changed slightly) are listed in Table 2.

As suggested by the reviewers, we also performed stochastic simulations based on our model to demonstrate that they reproduce observed features in the data. This is shown in the new Figure 7—figure supplement 2 and Figure 7—figure supplement 3, and discussed within the text. It should be noted that additional parameters would be required to incorporate XPD backsliding (and dissociation) observed at the end of the unwinding bursts. Therefore, simulations based on our model as currently constructed will *not* reproduce repetitive unwinding bursts. They will, however, generate individual bursts with the correct stepping dwell times at each position.